# Experiments on patterns of alluvial cover and bedrock erosion in a meandering channel

Roberto Fernández[1], Gary Parker[1, 2], Colin P. Stark[3]

[1]Ven Te Chow Hydrosystems Laboratory, Department of Civil and Environmental Engineering, University of Illinois at Urbana-Champaign, Urbana, IL 61801, USA
[2]Department of Geology, University of Illinois at Urbana-Champaign, Urbana, IL 61801, USA
[3]Lamont Doherty Earth Observatory, Columbia University, Palisades, NY 10964, USA

*Correspondence to*: Roberto Fernández (fernan25@illinois.edu)

**Abstract.**

In bedrock rivers, erosion by abrasion is driven by sediment particles that strike bare bedrock while traveling downstream with the flow. If the sediment particles settle and form an alluvial cover, this mode of erosion is impeded by the protection offered by the grains themselves. Channel erosion by abrasion is therefore related to the amount and pattern of alluvial cover, which are functions of sediment load and hydraulic conditions, and which in turn are functions of channel geometry, slope and sinuosity. This study presents the results of alluvial cover experiments conducted in a meandering channel flume of high fixed sinuosity. Maps of quasi-instantaneous alluvial cover were generated from time-lapse imaging of flows under a range of below-capacity bedload conditions. These maps were used to infer patterns of particle impact frequency and likely abrasion rates. Results from eight such experiments suggest that: (i) abrasion through sediment particle impacts is driven by fluctuations in alluvial cover due to the movement of freely-migrating bars; (ii) patterns of potential erosion are functions of sediment load and local curvature; (iii) low sediment supply ratios are associated with regions of potential erosion located closer to the inner bank, but this region moves toward the outer bank as sediment supply increases; and (iv) the threads of high erosion rates are located at the toe of the alluvial bars, just where the alluvial cover reaches an optimum for abrasion.

## 1 Introduction

In his report on the geology of the Henry Mountains, Gilbert (1877) advocated that the process of mechanical erosion of a bedrock river bed by material transported by the current depends on the hardness of the bedrock, the hardness, size and number of particles in transport, and the velocity of the stream. He noted that the number of sediment particles striking the bed and eroding it could increase up to the sediment transport capacity of the stream. At this point, the bed would be so crowded with particles that instead of colliding against the bed, they would collide against each other and the bedrock would be protected from erosion. Based on this observation, Gilbert (1877) stated that it is probable that the maximum work of mechanical erosion is performed when the load is far below the transport capacity of the stream.

During the last two decades, particular attention to the previously-described phenomenon has motivated experimental (e.g. Mishra et al., 2018; Hodge et al., 2016; Hodge and Hoey, 2016; Johnson and Whipple, 2010, 2007; Chatanantavet and Parker, 2008; Finnegan et al. 2007; Sklar and Dietrich, 1998), theoretical or numerical (e.g. Turowski 2018; Turowski and Hodge, 2017; Zhang et al. 2015; Inoue et al. 2014; Johnson 2014; Nelson et al. 2014; Nelson and Seminara, 2012; 2011; Lague, 2010;

Chatanantavet and Parker, 2009; Turowski et al., 2007; Whipple et al., 2000), and field (e.g. Ferguson et al., 2017; Beer et al. 2017, 2016; Beer and Turowski, 2015; Johnson and Finnegan, 2015; Inoue et al. 2014; Cook et al., 2013, 2009; Hodge et al., 2011) work examining the relation between sediment supply, degree of alluviation, and bedrock incision in mixed bedrock-alluvial rivers. Although Gilbert did not specifically used the terms 'tools' and 'cover' effects, he described them vividly. Saltating bedload particles in a bedrock river are one of the 'tools' needed to cause incision. As sediment supply increases to

a river reach, the ability to incise eventually decays due to the appearance of sediment deposits which protect the bed from further abrasion ('cover' effect). Therefore, in order for bedrock erosion to occur, a balance must exist between the 'cover' and 'tools' effects such that there are enough sediment particles in the system striking the bed, but not so many as to cover it and protect it from abrasion.

The experimental work of Sklar and Dietrich (2001) has led to a better understanding of the 'tools' and 'cover' effects. In their

work, the 'cover' effect was parameterized in terms of a cover factor $p_c$ which represents the areal fraction of bedrock that is covered by sediment. The exposed fraction is thus defined as $p_o = 1 − p_c$. The tools effect was parameterized as a linear dependence on sediment supply. The saltation-abrasion model of Sklar and Dietrich (2004) was the first to include these effects in a bedrock erosion model. The cover model used by Sklar and Dietrich (2006, 2004) to compute erosion linearly relates the areal fraction of the bed that is covered by sediment to the ratio of sediment supply to sediment transport capacity of a bed

fully covered with alluvium. The linear cover model has under certain conditions been validated via experimentation (e.g. Chatanantavet and Parker, 2008; Finnegan et al., 2007; Johnson and Whipple, 2010, 2007). Turowski et al. (2007) proposed an exponential cover model, which assumes that at below capacity transport conditions, sediment grains have equal probability of forming deposits over any part of the bed and cover could be static (immobile sediment) or dynamic (mobile sediment but still protects the bed from abrasion due to grain-grain interactions). Turowski and Rickenmann (2009), using a piezoelectric

bedload sensor, show some field evidence for the dynamic effect in the Pitzbach in Austria. Lague (2010) also proposed a bedrock channel morphodynamics model based on stochastic variations of discharge and sediment supply which accounts for alluvial thickness and its effect on limiting bedrock incision. His cover model is equivalent to the former two when working with the mean sediment thickness.

Recently Zhang et al. (2018, 2015) proposed the macro-roughness saltation-abrasion alluviation model which treats the cover

factor as the ratio between the alluvial thickness at a river cross section to the characteristic macro-roughness height of the bedrock surface. The advantages of this approach over the ones previously described is that by relating cover to alluvial thickness rather than sediment supply, it can deal with waves of alluviation and bed stripping and their dynamic effect on incision or the cessation thereof due to complete alluvial cover. Turowski (2018) presented a model that links alluvial cover to the width, slope and sinuosity of mixed bedrock alluvial rivers and postulates that change in channel width and sinuosity

over time depends only on the amount of alluvial bed cover. Finally, Mishra et al. (2018) conducted experiments in a U-shaped meandering channel with constant curvature and showed that: i) lateral erosion increases with sediment supply ratio, ii) vertical incision initially grows with sediment supply but shows a more complex relation due to the interplay between bedrock erosion and sediment deposition, and iii) zones of erosion along the toe of the point bar result in the formation of outer bedrock benches.

In spite of these developments, the cover factor definitions used so far by the different authors lack one or more important aspects required for the development of a model of bedrock incision in mixed bedrock-alluvial meandering rivers, namely:

(i) What are the roles of sediment supply and local curvature and how do they affect the areas of potential erosion in meandering bedrock-alluvial channels? With the exception of the recent work by Mishra et al. (2018), Inoue et al. (2017, 2016), Nelson et al. (2014), and Nelson and Seminara (2012), all models of bedrock incision by abrasion, or morphodynamics of mixed

bedrock-alluvial rivers are either '0D' or '1D' and most experiments have been conducted in straight channels (e.g. Johnson and Whipple, 2010, 2007; Chatanantavet and Parker, 2008; Finnegan et al., 2007). Even though Shepherd (1972) and Shepherd and Schumm (1974) did experiments with alluvial cover in bedrock analog substrates and report on erosion patterns in mixed bedrock-alluvial channels with some sinuosity, there is still no baseline set of experiments describing how the pattern of spatial cover is established in a meandering channel, and how it varies with local curvature and sediment supply.

(ii) What is the appropriate averaging window to characterize the areal fraction of alluvial cover? The model based on the areal fraction of cover uses an average value defined over an "appropriate" averaging window, but different definitions regarding its length scale and timescale have been provided to date. Moreover, this mean cover value assumes that the alluvial deposits covering the bed are transient. Field observations (Inoue et al. 2014, Cook et al. 2013, 2009) and laboratory experiments (Johnson and Whipple, 2010, 2007; Chatanantavet and Parker, 2008; Finnegan et al., 2007) indicate that zones of persistent

cover and persistent exposure coexist with transient deposits in mixed bedrock-alluvial rivers.

(iii) What is the role of alluvial cover fluctuations on erosion? Current models rely on a mean cover value, but temporal alluvial cover fluctuations provide a better representation of the frequency of the saltating bedload particle impacts on the bed which are responsible for bedrock erosion. Some authors have addressed this issue with probabilistic frameworks (e.g. Turowski and Hodge, 2017; Lague, 2010; Turowski, 2009) but physical measurements are still lacking and experimental data is required for

development and validation purposes.

(iv) What is the relation between alluvial cover and sediment supply? Most available models typically treat the relation between sediment supply and areal extent of alluvial cover by using a linear function. This relation has prevailed due to its simplicity and because it has been shown to be an acceptable approximation under certain circumstances (e.g. Chatanantavet and Parker, 2008). Nevertheless, there are models that predict a much wider range of behaviors (e.g. Turowski and Hodge, 2017; Hodge

and Hoey, 2012).

We addressed these questions by conducting experiments in a high-amplitude laboratory meandering flume to characterize the statistics of alluvial cover as they relate to the sediment supply ratio and local curvature. We also addressed the fourth question by conducting a simple experiment on a flat (non-sloping) bedrock slab. Before describing our experimental methods, we present the relevant definitions needed for our analysis.

## 1.1 Bedrock erosion and alluvial cover

The time rate of bedrock incision (erosion) by mechanical wear $E_s$ has been quantified with Eq. 1 by different authors (e.g. Sklar and Dietrich, 2004, 2006; Turowski et al., 2007; Chatanantavet and Parker, 2009) as follows;

$$E_s = V_i I_r p_o \tag{1}$$

In Eq. 1, $V_i$ is the volume of bedrock lost per particle impact, $I_r$ is the particle impact rate per unit area per unit time and $p_o$ is the fraction of exposed bedrock. The areal fraction of alluvial cover, i.e. cover factor $p_c$ is thus defined as $p_c = 1 - p_o$. A closely related equation to compute erosion is presented in Eq. 2 (e.g. Turowski et al., 2008; Chatanantavet and Parker, 2009). Let $\beta$ = a parameter that relates to the fraction of bedrock volume that is lost per particle impact at the end of each saltation, $q_{bt}$ - the capacity bedload transport rate per unit width for a bed fully covered with alluvium, and $q_{bs} = p_c q_{bt}$ - the actual bedload transport rate per unit width assuming that particles can only be mobilized from those portions of the bed that have an alluvial cover. Then.

$$E_s = \beta q_{bs} p_o = \beta q_{bt} p_c (1 - p_c) \tag{2}$$

But it is readily seen that the above relation breaks down for throughput load, bedload transport which enters and leaves a reach without depositing on the bed and does not contribute to cover in any meaningful sense. In Sklar and Dietrich (2004) and other works based on their cover model (e.g. Turowski et al., 2007; Lamb et al., 2008), $p_o$ represents the areal fraction of exposed bedrock and is related to the sediment supply ratio (Eq. 3). In the Zhang et al. (2015) cover model, $p_o$ represents the fraction of bed elevation at a given cross section which is not covered by alluvium, and is instead related to the ratio $\eta_a / L_{mr}$, where $\eta_a$ is a measure of the thickness of alluvium, and $L_{mr}$ is a measure of the intrinsic macro-roughness height of the bedrock surface itself (Eq. 4).

Both definitions are presented schematically in Figure 1. In general, $p_c$ is a function of $q_{bs}/q_{bt}$ or $\eta_a/L_{mr}$, and commonly used forms are given by Eq. 3 and Eq. 4.

$$p_c = 1 - p_o = \begin{cases} \frac{q_{bs}}{q_{bt}} & if \ 0 \leq \frac{q_{bs}}{q_{bt}} < 1 \\ 1 & if \ 1 \leq \frac{q_{bs}}{q_{bt}} \end{cases} \tag{3}$$

$$p_c = 1 - p_o = \begin{cases} \frac{\eta_a}{L_{mr}} & if \ 0 \leq \frac{\eta_a}{L_{mr}} < 1 \\ 1 & if \ 1 \leq \frac{\eta_a}{L_{mr}} \end{cases} \tag{4}$$

Eq. 3 or Eq. 4 in combination with Eq. 1 must be employed in terms of an appropriate averaging window over which to determine the cover fraction $p_c$ and open fraction $p_o = 1 - p_c$. For example, Sklar and Dietrich (2006), Gasparini et al. (2007) and Chatanantavet and Parker (2008) assume, explicitly or implicitly, that a) the averaging window is at least as large as channel width, and b) that $p_c$ fluctuates temporally between 0 and 1 within the window. If this were not the case, zones along the reach where $p_c$ persistently takes the values 0 and 1 would never be subject to incision, and channel geometry would not change over time in those reaches. However, if these assumptions are met over an appropriate time scale, all the reach would, in the long-term average, erode at the same rate (Sklar and Dietrich, 2004).

In the case of mixed bedrock-alluvial meandering rivers, where persistent alluvium deposits may form in e.g. point bars, the assumptions just described break down. In such rivers, erosion occurs only in areas with transient cover and is not expected to occur in areas that are persistently covered or exposed. Under certain conditions, specific areas of the channel might have little to no probability of being struck by sediment particles, thus limiting the areas that could undergo erosion.

## 2 Materials and Methods

### 2.1 Flume

Experiments were conducted in the Kinoshita Flume at the Ven Te Chow Hydrosystems Laboratory, University of Illinois at Urbana-Champaign. The flume, shown in Fig. 2b, is 0.60 m wide, 0.40 m deep and 33 m long (along the centerline, not including upstream and downstream tanks), and has a sinuosity of 3.7. All three meander bends are identical and have a down-channel wavelength of 10 m. All results presented herein correspond to experiments conducted with water flowing from right to left as indicated in Fig. 2c, i.e. with the bends skewed in the upstream direction. The flume is a closed system in which water and sediment are recirculated. Readers interested in more specific details about the Kinoshita flume are referred to Abad and Garcia (2009a, b).

### 2.2 Bed-material properties and bed characteristics

The alluvium used in the experiments was crushed walnut shells, which have a specific gravity in the range 1.3-1.4. A bedrock basement was built in the flume using the bathymetry measured by Czapiga (2013), who conducted experiments under fully alluvial conditions using the crushed walnut shells. The supplemental material (S3) shows the bathymetry from those experiments. Transverse slopes were measured from the bathymetry, and a relation between transverse slope and streamwise location was fit to the data. Using it, transverse slopes every 0.5 m were calculated. Based on the computed transverse slopes, cross-sectional bathymetric slices were cut out of foam and placed inside the flume every 0.5 m. Pea gravel was used to fill the flume following the profile established by the foam slices. The region between streamwise stations CS07 and CS23 (Fig. 2c) was filled with gravel to an elevation slightly below the maximum given by the foam slices. This section was then covered with a ~ 1 cm layer of concrete and used to create the bedrock surface. We filled the rest of the flume (CS00 – CS07, and CS23 – CS30) with pea gravel to maintain an average centerline elevation throughout the flume. The size of the gravel was chosen so as to prevent it from being transported by the flow in the experiments. Figure 3c shows the bedrock bed built inside the Kinoshita Flume, and Fig. 2a shows its bathymetry. The concrete was painted white to enhance the contrast between the bedrock and the alluvium. The supplemental material (S1, S3) has a set of images and diagrams which provide additional information regarding the construction of the bedrock bed inside the flume and the pre-mixed concrete used.

The grain size distributions of the crushed walnut shells, the pea gravel, and the dry concrete mix (including gravel, sand and cement) are shown in Figure 3a. The inset figure includes the results of laser scans conducted to measure the as-built bedrock macro-roughness, i.e. the difference between the maximum and minimum elevations according to Zhang et al. (2015).

## 2.3 Bed laser scans

A Keyence LB-1201 laser (Keyence, 1992) with sub-millimeter precision (250 µm) was used to scan the bed at five different locations, namely: CS10, CS12, CS15, CS17 and CS20 (Fig. 2c). These locations were chosen because they are representative of the bedrock topography at the apices (CS10, CS15 and CS20) and the crossings (CS12 and CS17). A polynomial was fit to the scans, and residual elevations were calculated by subtracting the actual reading from the polynomial. This removed the local topography from the signal. The average residual elevation along the cross sections was calculated and used to estimate the macro-roughness of the bedrock bed. The resulting value (10 mm) is also indicated in Fig. 3a. More details about the scans and the polynomial fit are included in the supplemental material (S1).

## 2.4 Areal alluvial cover measurements

The percentage of areal alluvial cover was calculated by analyzing time-lapse images of the flume bed. Images were taken, on average, every 10 s (0.1 Hz) during the duration of every run and processed in Matlab. A region of interest (ROI) was selected for each image series. In this study, the ROI corresponds to the middle bend of the Kinoshita flume, i.e. between streamwise locations 10 m and 20 m (Fig. 2).

Images were first converted to gray scale, and then the method of Otsu (1979), as implemented in Matlab ('graythresh' function), was used to make the images binary. The resulting black (alluvial cover) and white (bedrock) images were used to calculate the percent areal cover. The fraction of alluvial cover was determined as shown in Eq. 5.

$$p_{c_{ROI}} = \frac{\left(N - \sum_{j=1}^{N} px_j\right)}{N} \tag{5}$$

In Eq. (5), $pc_{ROI}$ = percent of areal alluvial cover inside the region of interest; $N$ = total number of pixels inside the region of interest (i.e. total area); and $px_j$ = value of the $j^{th}$ pixel in the binary image (white pixels are equal to one and black pixels are equal to zero). The pixel size in the images was approximately 2.8mm x 2.8mm. This resolution is not enough to capture individual sediment grains ($D_{50}$ ~ 1.6 mm). Nevertheless, if a single grain lies in a pixel, ~ 25% of its area would be alluvium, leading to a gray pixel (as seen by MatLab). Depending on the threshold determined by Otsu's method for the region of interest, a single grain could be enough to be interpreted as alluvial cover. More details regarding the image acquisition and processing are included in the supplemental material (S2).

## 2.5 Relation between alluvial cover and sediment supply

A rectangular bedrock slab was built with the same materials used to build the bedrock basement in the flume. The bedrock slab was built over a piece of foam laid on a floor so as to have no longitudinal or transverse slope. Pea gravel was placed over the foam and a thin layer of concrete was poured over it. It was then painted white to increase contrast between the bedrock and the alluvium. The purpose of this bedrock slab, which was 0.6 m long by 0.4 m wide, was to measure i) the relation between areal alluvial cover and sediment mass fraction, and ii) the relation between areal alluvial cover and the ratio of

alluvial cover thickness to bedrock macro-roughness. Images of this simple experiment are included in the supplemental material (S1).

To quantify the cumulative sediment mass fraction, known weights of sediment were incrementally added to the slab and spread evenly until the bed was fully covered with alluvium. Eleven iterations were necessary to cover the bed completely. The total amount of mass used was 646 g. Mass increments used in every iteration are shown in supplement S1. The cumulative sediment mass fraction was calculated as the cumulative weight of sediment in every iteration, divided by the total weight of sediment used to fully cover the bed. Areal alluvial cover was quantified using images, and following the approach described in section 2.4. The ratio of alluvial cover thickness to bedrock macro-roughness was quantified by scanning nine cross sections of the bedrock slab with a sub-millimeter precision Keyence laser (Keyence, 1992). The cross sections were 4 cm away from each other. The first set of scans were conducted over the bare bedrock slab, and then they were repeated each time that alluvium was added over the slab. The entire process was conducted two times to verify that the results would not change due to any human-induced errors in the measurements or the way in which the alluvium was distributed over the bed after each iteration. After the first set of measurements, the alluvium was initially removed with a brush and then with an air-pressure hose to make sure no grains were left on the slab. Images related to this experiment are included in the supplemental material (S1).

## 2.6 Experimental conditions

Table 1 shows the general experimental conditions used in this study. The flow discharge rate used in all runs was 12.3 liters per second (Ls$^{-1}$) which corresponds to the flow rate used by Czapiga (2013). This flow rate created the alluvial bathymetry used to build the bedrock bed in these experiments. The flow discharge was measured with electromagnetic flow meters. Given that the sediment recirculating pump only works at a constant discharge of 3.1 Ls$^{-1}$, the main pump was set to have a discharge of 9.2 Ls$^{-1}$.

The volume of sediment inside the Kinoshita flume was modified between runs so as to obtain different reach-averaged areal ratios of alluvial cover. Runs in this study are identified based on this value (Table 2). For example, run "pc79" had 79% of the total bed area covered with alluvium after averaging in space (one wavelength) and time (one hour). The first run conducted was pc79. After this run, we removed sediment from the flume, leading to lower pc conditions. The values obtained were not planned. The two following runs were pc72 and pc54. Afterwards, all the sediment was removed from the system to run the bare-bedrock condition, pc00. The following runs were pc19, pc27, pc38 and pc46, conditions that were achieved after progressively adding sediment to the flume.

We allowed the bed to adjust for at least 8 hours between runs. The 60 minutes we report in the study are after the bed had adapted the new condition. We computed the alluvial cover statistics throughout the transition from one state to another and once it had reached equilibrium we continued measuring. We report only the values once the system had reached equilibrium for each condition. Water surface slopes were initially calculated by using the water level

elevation changes in the upstream and downstream tanks of the Kinoshita flume. Both tanks have a measuring tape glued to the upstream- and downstream-most walls (Fig. 2b). These measuring tapes were used to guarantee that runs always started at the desired water elevation. Before turning on the pumps, desired water elevations were verified, and after the run had started, readings were taken every 20-30 minutes.

Water surface elevations were also measured with eTapes in runs pc00, pc19, and pc79 (Fernández, 2018). An eTape is a sensor with a resistive output that varies with the level of fluid in which it is immersed. The resistive output of the sensor is inversely proportional to the height of the water. Low water depths correspond to high output resistance. Conversely, high water depths correspond to low output resistance. Details about the eTape installation, calibration, and operation are given in S4, and further information may be found in Fernández (2018). After runs pc79, pc72 and pc54 were finished, we noticed that

the water surface slopes in the Kinoshita flume were different depending on if they were calculated for the total length of the flume, i.e. between tanks, or for the middle bend of the flume only. Figure 4 shows an example of the water surface elevations measured with the eTapes (middle zone of flume) and the measuring tapes (entire flume) for run pc79. To accurately measure the middle bend water surface slopes in runs pc00-pc46, point gages were placed on the flume at streamwise locations 9 m and 21 m (Fig. 2). The slopes calculated with the point gage readings are shown in Table 1. The average ratio of the slopes

calculated with the point gages to those calculated with tank elevations in runs pc00 – pc46 was used to estimate the slopes in the middle bend of the flume for runs pc54, pc72 and pc79.

The sediment transport rates were measured by collecting material in a box fitted to the diffuser at the upstream end of the flume (Abad and Garcia, 2009b). This box is not shown in Figure 2. Table 2 shows the average sediment transport rates measured.

**2.6 Quantifying erosion potential**

Based on Eq. 2, a dimensionless erosion potential $E_{sp}$ may be expressed as a function of the areal fraction of alluvial cover as shown in Eq. 6 below. We use this (Sec. 3.5) to assess the spatio-temporal average erosion potential for the seven experimental conditions with alluvium.

$$E_{sp} = p_c(1 - p_c) \qquad (6)$$

At the microscopic level, the value of $p_c$ can only take values of zero (exposed bedrock) or one (covered with alluvium). In the context of the areal images obtained during the experiments, this means that pixels may change between white and black throughout the run. This information may be used to quantify erosion potential based on alluvial cover fluctuations.

Bedrock incision can only occur when a particle strikes the bed. If a pixel changes from white to black between consecutive

images, it means that sediment particles traveled into the area and struck the bed. If the pixel remains black or white in consecutive images, no strikes occurred; and if the pixel changes from black to white, sediment particles have left, and thus did not strike the bed. With these definitions, the erosion potential may be quantified by counting the number of times that a pixel changes from white to black, i.e. by quantifying the fluctuations in alluvial cover.

The frequency of strikes ($f_s$) at the $j^{th}$ pixel corresponds to the number of times that the $j^{th}$ pixel has changed from white ($p_c = 0$) to black ($p_c = 1$) between consecutive images (im), divided by the total number of images (N) in the series (Eq. 7). We use this approach (Sec. 3.6) to assess the erosion potential based on alluvial cover fluctuations for the seven experimental runs containing alluvium.

$$f_{s_j} = \frac{\sum_{i=1}^{N}\left(\frac{dp_C}{dim_i}=-1\right)_j}{N} \tag{7}$$

## 3 Results

### 3.1 Relation between alluvial cover and sediment supply

Figure 5 shows the relation between alluvial cover and sediment supply measured on the bedrock slab and in the Kinoshita flume. Specifically, Figure 5a shows the relation between areal alluvial cover and cumulative sediment mass fraction measured on the bedrock slab. Figure 5b shows the relation between areal alluvial cover and the ratio of alluvial cover thickness to bedrock macro-roughness measured on the bedrock slab. The thin dashed lines with circle and square markers show the average results of the measurements; the thick dashed lines correspond to a best-fit line; and the dotted lines show the linear relation between variables that has been used by previous authors (e.g. Zhang et al., 2018, 2015; Inoue et al., 2016, 2014; Chatanantavet and Parker, 2009, 2008; Sklar and Dietrich, 2006, 2004). Figure 5c shows the relation between reach-averaged alluvial cover (spatial average measured over one wavelength) and sediment supply ratio measured in the Kinoshita flume.

The relations between alluvial cover and sediment mass fraction in the bedrock slab (Fig. 5a) and the Kinoshita flume (Fig. 5c) are logarithmic (Eq. 8). The value of the constant 'a' in Eq. (8) below is different between the bedrock slab (a =0.23) and the Kinoshita flume (a = 0.14), but the shape of the relation is the same. Previous research has shown that different relations between percent cover and sediment supply ratio are valid under certain circumstances (e.g. Turowski and Hodge, 2017; Inoue et al, 2014; Chatanantavet and Parker, 2008) and our results suggest that the relation below is also possible:

$$p_c = a \cdot ln\left(\frac{q_{bs}}{q_{bt}}\right) + b \tag{8}$$

This relation diverges as the sediment supply ratio (term in parentheses) tends towards zero and as such, does not describe the small sediment flux limit. The relation is similar to those derived by Turowski and Hodge (2017, Eqs. 27 and 31) when the exponential term in their relation is small. Aubert et al. (2016) predict a similar relation to describe the alluvial cover based on direct numerical simulations. In the case of the bedrock slab, the logarithmic relation suggests that, initially, the areal cover increases rapidly with sediment supply ratio. Once the smaller voids in the bed are filled, more and more alluvium is needed to fully cover the largest roughness elements and further increase $p_c$.

In the case of the Kinoshita flume, the logarithmic relation between alluvial cover and sediment supply ratio is believed to be due in large part to the formation of point bars and transient alluvial deposits. Initially, a small amount of alluvium covers a

proportionately larger area of the bed, but as sediment supply increases, alluvial thickness growth is favored over areal extent of alluvial cover. As more alluvium accumulates over regions previously covered, additional sediment supplied to the reach tends to deposit at the edge of the existing deposits, thus increasing alluvial cover, but at an ever smaller rate.

Figure 5b shows the relation between areal alluvial cover and alluvial thickness to bedrock macro-roughness ratio. Zhang et al. (2018, 2015) and Inoue et al. (2014) used the assumption that the relation is linear but the results obtained for the bedrock slab suggest that an 'S-shaped' (sigmoid curve) relation is more appropriate. A logistic curve, which is a type of sigmoid curve, was fit to the measurements in this study. Eq. 9 shows the general logistic function and Eq. 10 shows the one used here. Comparing the two, it may be seen that: $x = \eta_a/L_{mr}$; $f(x) = p_c(\eta_a/L_{mr})$; $L$ is the maximum value of the curve, corresponding to $p_{c\,max} = 1.0$; $k$ is the steepness of the curve; and $x_o$ is the x-value of the sigmoid curve's midpoint. As shown in Figure 5b and Eq. 10, the steepness used to fit the sigmoid curve to the measured values was 8 and the midpoint was defined at $\eta_a/L_{mr} = 0.4$.

$$f(x) = \frac{L}{1+e^{-k(x-x_o)}} \tag{9}$$

$$p_c(\eta_a/L_{mr}) = \frac{1.0}{1+e^{-8[(\eta_a/L_{mr})-0.4]}} \tag{10}$$

The function in Eq. 10 is valid between a characteristically low (e.g. 0.05) and a characteristically high (e.g. 0.95) value of $\eta_a/L_{mr}$ to avoid unrealistic cover values (Zhang et al. 2018). It is likely that the steepness and midpoint value are associated with some measure of the grain size distribution of the alluvium and the macro-roughness height of the bedrock. In the case of the bedrock and alluvium (Fig. 3a) used in this study, the steepness value corresponds to $k \sim L_{mr}/D_{16}$ and the mid-point value corresponds to $x_o \sim 2.1 D_{84}/L_{mr}$. This issue merits further investigation so as to define appropriate relations to calculate the steepness and mid-point value of the sigmoid curve for implementation in numerical models. We discuss the issue of alluvial thickness and alluvial cover further in section 4.4.

Figure 6 shows a snapshot of the middle bend of the Kinoshita flume corresponding to each one of the eight reach-averaged alluvial cover conditions. Similar images for the bedrock slab experiment are included in the supplemental material (S1). Links to the videos showing the bed evolution for the different experimental conditions are included in the 'Video supplement' section at the end of the paper.

**3.2 Reach averages of alluvial cover fraction**

Figure 7 shows the temporal evolution of reach-averaged alluvial cover for all experimental runs. Larger fractions of alluvial cover are associated with fluctuations about the mean value due to the appearance of freely-migrating bars as sediment supply increases. Figure 8 shows the maps of alluvial cover for all experimental runs. Darker shades of blue correspond to areas that were covered with alluvium for more than 70% of the time; and shades of yellow correspond to areas that were covered with alluvium less than 30% of the time. In regards to the 'tools' and 'cover' effects, the white and black regions in those alluvial

cover maps would not experience erosion. No tools (alluvium) are available to erode the bed in the white regions whereas alluvium completely covered the bed in the black regions, thus protecting it from erosion.

The areal alluvial cover definition in Fig 1a is based on the assumption that alluvial deposits are transient, i.e. no portions of the bed in the reach remain persistently covered with alluvium or fully exposed. This assumption is not met in meandering channels where persistent alluvial cover deposits form and grow as sediment supply increases, and erosion may only occur in those regions where alluvial cover is changing in time, i.e. regions with transient cover.

### 3.3 Regions with transient alluvial cover

The alluvial cover maps in Figure 8 show different percentages of persistently covered or exposed bedrock, as well as regions with transient alluvial cover. The regions with transient alluvial deposits are those over which alluvial cover is changing in time (colored regions in Fig. 8). To delineate and quantify these areas, the following criteria were used: regions with persistent alluvial cover are those in which $p_c > 0.975$; regions with persistent exposed bedrock are those in which $p_c < 0.025$; and regions with transient alluvial cover are those in which $0.025 \leq p_c \leq 0.975$. Using these criteria, maps of transient alluvial cover were prepared. Figure 9 shows the regions of transient cover (gray), persistent cover (black) and persistently exposed bedrock (white) for each of the eight experimental conditions. The area of the former two regions increases with sediment supply, whereas the area of the latter decreases as sediment supply increases.

Figure 10 shows the reach-averaged percentages of these three regions for all eight experimental conditions. Therein, the yellow dashed line corresponds to the reach-averaged fraction of persistently exposed bedrock; the blue line corresponds to the reach-averaged fraction of persistently covered bedrock; the light-blue line corresponds to the reach-averaged fraction of the bed with transient cover; the thick black line corresponds to the sum of the transient and persistent cover fractions; and the black dotted line corresponds to the 1:1 line.

The regions of persistent and transient cover increase as a function of reach-averaged alluvial cover. The regions of persistently exposed bedrock decrease concomitantly. In general, both the fraction of the total area with persistent and transient cover grow at a similar rate with increasing reach-averaged $p_c$. The reach-averaged conditions for which transient and persistent cover have similar area ratios are $p_c = 0.27$, $0.46$, $0.54$ and $0.72$. The largest differences between persistent and transient cover are observed at $p_c = 0.19$, $0.38$ and $0.79$.

The case $p_c = 0.19$ is likely due to the typical sedimentation patterns observed in meandering bedrock channels when alluvial point bars first form. Immediately downstream of the bend apices, i.e. points of highest curvature, sediment is deposited. In the Kinoshita flume, the apices of bends are located at streamwise locations 9.5 m, 14.5 m and 19.5 m (Fig. 13). Initially, these locations become the upstream-most points of the point bars. Once these deposits have been established, and as long as sediment continues to be supplied from upstream, the incoming particles travel above the existing deposit due to decreased resistance from the bed. Under such conditions, persistent alluvial cover is favored over transient alluvial cover.

The case $p_c = 0.38$ has a larger portion of the total area with transient cover than with persistent cover. As more sediment was supplied to the system while keeping the initial (no flow) water depth constant (Table 1), the alluvial thickness could not

continue to grow indefinitely but rather, the areal extent of alluvial cover grew instead and the water-surface slope also increased (Table 2). Sediment particles could no longer be preferentially transported over the alluvial deposits, and began to be transported closer to the edge of the existing deposits.

The case $p_c = 0.79$ shows a dip in the ratio of transient cover, while the area with persistent cover continues to increase. Although there are no runs with a larger reach-averaged fraction of alluvial cover, it is likely that this trend would be maintained until the bed is completely covered with alluvium. As $p_c$ grows, the area ratio of persistently covered regions should increase at a faster rate, and the area ratio of regions with transient cover should decrease rapidly towards zero. Eventually, the channel will not have any area left for the areal extent of alluvial cover to grow, so that further deposition promotes increased alluvial thickness instead. Erosion by abrasion would promote lateral migration of the bedrock river (e.g. Inoue et al., 2017; Shepherd, 1972).

## 3.4 Cross-section averages of alluvial cover

Figure 11 shows the cross-sectional alluvial cover averages for the seven experimental conditions with alluvium. Values were extracted every meter between streamwise locations 10 m and 20 m. Therefore, eleven local alluvial cover values were obtained for each experiment. As in the case of the reach-averaged values, these results include persistently covered and exposed portions of the cross section as well as a fraction with transient alluvial cover.

In general, all conditions exhibit similar trends, with local lows in $p_c$ at streamwise locations 15 m and 19 m and local highs at streamwise locations 11 m and 16 m. The regions showing higher local percentages of alluvial cover are located 1.5 m downstream of the bend apices. Point bar deposits are responsible for the higher local value of $p_c$ at these locations. On the other hand, the local lows in $p_c$ are associated with the points of highest curvature in the reach. Both local lows are within 0.5 m of the bend apices.

Figure 12 shows the ratios of the cross sections that had persistently exposed bedrock (dashed yellow line), persistently covered bedrock (blue line), transient alluvial cover (light-blue line), and the ratio corresponding to the sum of persistent plus transient cover (black line) for all experimental conditions but pc00. The ratio of exposed bedrock peaks in the vicinity of the bend apices. Even in the case of reach-averaged $p_c = 0.79$, portions of the bed in these areas remain exposed due to high curvature. Except for the cases with reach-averaged $p_c = 0.38$ and 0.54, no cross sections have fractions with transient alluvial cover greater than 60%.

The average fractions of transient alluvial cover at the cross-sectional level have values of 0.10 for $p_c = 0.19$, 0.21 for $p_c = 0.27$ and between 0.31 and 0.34 for the other experimental conditions. In spite of the local variations in transient alluvial cover, potential erosion is, on average, limited to a rather small portion of the cross section. This is likely due to the combined effects of sediment supply ratio and local curvature.

Figure 13 shows box plots of cross-sectionally averaged $p_c$ normalized with the reach-averaged value. The figure also shows the dimensionless curvature of the Kinoshita flume (black dashed line), the negative value of the curvature (gray dotted line)

and the median normalized values of $p_c$ (red line). The true ($\kappa$) and negative ($-\kappa$) centerline curvature signals are shown to better highlight the trend of normalized $p_c$ with curvature.

The boxes include information from the seven experiments at each cross section. The median value is indicated by the red line inside the box; the bottom line on each box corresponds to the first quartile ($q_1$); the top line on each box corresponds to the

third quartile ($q_3$); whiskers extend to $q_1 - 1.5(q_3 - q_1)$ at the bottom and $q_3 + 1.5(q_3 - q_1)$ at the top; and values lying outside this range are considered outliers and are indicated with a red cross. The cross sections located close to the bend apices, i.e. regions with local high curvature, show normalized $p_c$ values below unity, whereas the regions with smaller curvature values show normalized $p_c$ values above unity. Normalized, local $p_c$ values follow the overall trend of local curvature.

**3.5 Erosion potential based on alluvial cover averages**

Figure 14 shows the erosion potential (Eq. 6) for all experimental conditions. Regions with higher erosion potential are those for which alluvial cover averages were close to 0.5, in accordance with the parabolic form of Eq. 6. These regions are shown in dark blue in figure 13. White regions have no erosion potential due to a lack of tools or the presence of alluvial cover protecting the bed from abrasion. The regions of potential erosion are limited to the areas with transient alluvial cover. In

general, their width is a function of sediment supply ratio, with narrower regions associated with smaller sediment supply ratios. Locally, the width of these regions is affected by curvature as well, with narrower regions in areas of high curvature, and wider regions in areas of lower curvature.

The region of potential erosion is located closer to the inner bank for lower sediment supply ratios, and moves outward as sediment supply increases. Focusing on the region of potential erosion located at the bend apex at streamwise location 14.5 m

(see Fig. 2c for location on plots), it is seen that for $p_c = 0.27$, the region is located right next to the inside bank whereas for $p_c = 0.79$, the region is much closer to the outer bank.

**3.6 Erosion potential based on alluvial cover fluctuations**

The results of alluvial cover shown and discussed up to this point correspond with spatial or temporal averages. Nonetheless,

Fig. 15 shows the frequency of strikes ($f_s$) for all experimental conditions (Eq. 7). In general, the areas in color in the figure are similar to the areas with transient alluvial cover shown in Fig. 9 and the areas with erosion potential in Fig. 14. Picking out differences in these particular figures is not straightforward, but the videos included in the supplemental material illustrate the migrating erosion fronts, and suggest that erosion is likely to be driven predominantly by the movement of freely-migrating bars. The use of the frequency of strikes associated with fluctuations in alluvial cover provides an improved approach for

computing bedrock erosion by abrasion, as discussed below.

## 4 Discussion

### 4.1 Transient alluvial cover: An issue of timescales

Alluvial deposits on the bed of a bedrock river cover it and protect it from abrasion (Gilbert, 1877; Sklar and Dietrich, 1998; 2004). At the microscopic level, a portion of the river bed can only be covered ($p_c = 1$) or exposed ($p_c = 0$) in any given instant.

Therefore, a notion of transient alluvial deposits becomes necessary to guarantee that in time, $p_c$ fluctuates between those end members and erosion caused by saltating bedload particles is possible (Eq. 2). In the original saltation-abrasion framework of Sklar and Dietrich (2004), transient alluvial deposits were an underlying assumption (Fig. 1a). No temporal averaging window was needed since all portions of the bed had the same probability of being eroded in time.

Turowski et al. (2007), working under the assumption that sediment transport capacity is uniform across a control area of

unspecified dimensions, described the cover effect as static or dynamic. Static cover occurs when the amount of sediment supplied to the reach is larger than the transport capacity, and therefore, some particles remain immobile on the bed. Dynamic cover occurs when the amount of sediment supplied is smaller than the transport capacity. Sediment particles cover portions of the bed but are mobile. As sediment supply increases, erosion is limited due to more grain-grain collisions than grain-bed interactions. In this framework, the dynamic cover effect reduces erosion by decreasing the impact energy experienced by the

bed due to the interactions between grains. Nonetheless, the notion of transient alluvial cover over an unspecified time-window is required, and in the long term, all areas of the bed have the same likelihood of being eroded as long as sediment supply is below the transport capacity of the reach.

Chatanantavet and Parker (2008), Inoue et al. (2014), Hodge and Hoey (2016b), and Ferguson et al. (2017) present cases in which sediment particles are being transported over the bed as throughput bedload. These cases challenge the notion of a cover

effect because alluvial deposits do not exist at all. They could still be treated as having transient alluvial cover for modeling purposes but what would be the relevant timescale to characterize it? Chatanantavet and Parker (2008), and Hodge and Hoey (2016b) also observed that throughput load may be unstable; as soon as hydraulic conditions change, runaway alluviation occurred and the same portion of the bed changed from a state of being continuously struck by sediment particles (undergoing erosion) to being protected from further erosion.

The examples above suggest that areas with persistent or transient alluvial cover in a mixed bedrock-alluvial river can only be categorized as such given a specified timescale. The reach-averaged results shown in Fig. 8, Fig. 9 and Fig. 10 suggest that the areas subject to erosion in mixed bedrock-alluvial meandering rivers are a fraction of the total reach area. In this study, we defined transient alluvial cover as those portions of the bed where temporal averages of local alluvial cover had values between $0.025 < p_c < 0.975$ during the time of the experiment (Fig. 8). Based on this definition, areas with persistent alluvial cover or

exposed bedrock were also delimited (Fig. 9). In the case of the Kinoshita flume experiments, the areas with transient alluvial deposits occupied less than 50% of the total reach area, hence erosion could only occur within a restricted portion of total bed area.

The problem remains in regards to generalizing appropriate timescales for modeling purposes. Our results are based on a constant discharge but in real rivers, a flood could mobilize all alluvium on the bed of the channel and within the timescale of the flood, alluvial cover would also be transient (e.g. Turowski and Rickenmann, 2009). The use of temporal averages of alluvial cover has limitations, and our results suggest that characterizing the fluctuations of alluvial cover may be a better approach.

## 4.2 Alluvial cover fluctuations vs. averages

Figure 16 shows a hypothetical example of two cases in which the long-term average of alluvial cover is equal, but the fluctuations in alluvial cover between them are different. Given that erosion by abrasion is driven by the number of times the bed is struck by particles, erosion would only occur in the first case. Erosion would only occur each time the area changes from white to black, i.e. every time a particle moves into the area and strikes the bed upon arrival. This simple example suggests that the use of temporal averages of alluvial cover to calculate erosion may lead to inaccurate results.

The use of a relation such as Eq. 2 with spatiotemporal averages of alluvial cover also has limitations. According to it, the following experiment pairs: i) $p_c = 0.19$ and $p_c = 0.79$; ii) $p_c = 0.27$ and $p_c = 0.72$; and iii) $p_c = 0.46$ and $p_c = 0.54$ should have very similar, or equal, erosion potentials (Eq. 6) as shown below:

i) $\quad E_{sp} = 0.19(1 - 0.19) = 0.19(0.81) = 0.154 \quad and \quad E_{sp} = 0.79(1 - 0.79) = 0.79(0.21) = 0.166$

ii) $\quad E_{sp} = 0.27(1 - 0.27) = 0.27(0.73) = 0.197 \quad and \quad E_{sp} = 0.72(1 - 0.72) = 0.72(0.28) = 0.202$

iii) $\quad E_{sp} = 0.46(1 - 0.46) = 0.46(0.54) = 0.248 \quad and \quad E_{sp} = 0.54(1 - 0.54) = 0.54(0.46) = 0.248$

Figure 14 and the videos in the supplement show that the erosion potential in all cases is different, thus suggesting that spatial averaging may also lead to inaccurate results. For these reasons, temporal and spatial averages of alluvial cover are not appropriate to quantify erosion in mixed bedrock-alluvial rivers. The computational method of Inoue et al. (2016, 2017) both tracks the migration of cover fronts and bars and calculates cover at a spatiotemporally local level, thus approaching the methodology suggested here.

## 4.3 Regions of preferential erosion in mixed bedrock-alluvial meandering rivers

In spite of characterizing erosion potential with spatio-temporal averages (Fig. 14) or fluctuations (Fig. 15) of alluvial cover, the regions of preferential erosion in our experiments show some characteristics that are worth discussing. In all experiments, the regions of preferential erosion are located at the edges of persistent alluvial cover deposits. Their precise location and width are a function of sediment supply and local curvature.

In general, as sediment supply increases, the areas of preferential erosion moved outwards. Our results suggest that in all cases, inset channels would have been formed at the edge of alluvial deposits and bank erosion would have only occurred beginning

at CS11 (Fig. 2c, 15) for pc54, pc72 and pc79. Downstream of CS15, outer bank erosion would only occur for the cases with pc72 and pc79. Therefore, higher sediment supply is needed for bank erosion to occur, and under low sediment supply, inset channels and outer bedrock benches, are likely to form. Figure 17 shows an image of the mixed bedrock-alluvial Shimanto River in Shikoku, Japan and a sketch of what the cross section might look like with the areas of erosion and no erosion

indicated. The reach shown in the image has an alluvial point bar on the inside of the bend, a narrow inset channel at the edge of the point bar and an exposed bedrock bench on the outside of the bend. The same morphologies have been observed in a smaller scale stream called Pescadero Creek, in California, USA (Fig. 6B in Johnson and Finnegan, 2015). The experiments of Mishra et al. (2018) also show that when sediment supply is low, the alluvial point bar is narrow and an inset channel is eroded at the toe of the point bar, leaving an exposed bedrock bench on the outer part of the bend.

The typical geometry of an alluvial meandering channel cross section is shallow on the inside and deep on the outside. The reach of the Shimanto River shown in Figure 17 has a different geometry. The deepest portion of the channel is not located on the outer bank. Instead, it is located at the toe of the point bar, which happens to be approximately at the middle of the cross section. It is likely that the narrow inset channel was formed during a long period of decreased sediment supply. During this period, the region of transient alluvial cover was confined to the current width of the channel shown in the image. The outer

bedrock bench could potentially be eroded if sediment supplied to the reach from upstream were to be increased, and maintained at this increased value for an extended period of time. If this occurred, the point bar would likely extend toward the outer part of the bend, thus moving the area of transient alluvial cover farther into this region. Finnegan et al. (2007) observed similar trends in experiments conducted in a straight flume over an erodible bed, where erosion began at the edges of sediment patches and formed longitudinal grooves in the channel. Shepherd and Schumm (1974) also observed that outward

bank erosion was possible when bed material was transported at capacity, but when the amount of material in transport was less than the transport capacity, inset channels formed. Similar observations have been made experimentally by Mishra et al. (2018) and numerically by Inoue et al. (2017), and Nelson and Seminara (2011). Even though we did not measure velocities, our observations suggest that the areas with very narrow regions of erosion potential, e.g. between CS14 and CS15, are located at regions of topographically induced high flow velocities in accordance with observations made by Hodge and Hoey (2016b).

The specific links between sediment supply and local curvature, even though suggested by our results, need further investigation to properly parameterize them. It is likely that antecedent curvature and curvature sign also play a role (Fig. 13). Moreover, the use of denser material, e.g. sand, would likely affect the specific locations of the alluvial deposits. However, the main trends observed herein are likely to be general.

## 4.4 Alluvial thickness and alluvial cover

The results corresponding to the Kinoshita flume are based on areal cover of alluvial sediment captured with a camera located above the flume. These observations are related to the framework of Sklar and Dietrich (2004) described in Fig. 1a. The framework proposed by Zhang et al. (2018, 2015) and shown in Fig. 2b, relates to the experiment conducted in the small bedrock slab (S1, Fig. 5) where the cover is quantified as in Eq. 4. That experiment allowed us to relate the areal cover fraction

to the ratio of alluvial thickness to bedrock macro-roughness (Fig. 5b). We obtained and S-shaped relation between these two variables.

This result provides a useful link between the two models (Fig. 1a and 1b) but is only constrained by geometric variables, specifically the alluvium grain size and the bedrock macro-roughness. Other factors that affect the distribution and size of alluvial deposits are: local topography and hydraulic conditions (e.g. Hodge and Hoey, 2016b; Chatanantavet and Parker, 2008; Finnegan et al., 2007; Johnson and Whipple, 2007), grain size and ratio between grain and bedrock roughness (e.g. Ferguson et al., 2017; Nelson et al. 2014; Johnson 2014; Inoue et al. 2014; Chatanantavet and Parker, 2008), feedbacks between bedrock erosion, sediment deposition and its effects on hydraulic resistance (e.g. Ferguson et al. 2017; Nelson et al. 2014; Johnson, 2014; Inoue et al. 2014) and channel sinuosity (Shepherd and Schumm 1974; Shepherd, 1972).

The relations between amount of sediment in the system and alluvial cover in Fig. 5a and Fig. 5c are similar but are not equivalent. Figure 5a is based on the cumulative mass of sediment added to the bedrock slab, whereas Fig. 5c is based on the sediment supply ratio (Eq. 3). Turowski and Hodge (2017) developed an equation to relate the sediment mass on the bed and sediment supply. Their model differentiates between the mass of mobile and stationary bed material and relates them to sediment flux via an entrainment-deposition equation. Their framework could be tested with our dataset. Two particular issues of interest are:

    i)    The area of exposed bedrock is a function of the sediment mass in the system and the probability of incoming particles striking open bed areas. Turowski and Bloem (2015) showed that particle impact energy can be transferred to the bed if the thickness of the alluvial layer, even if static, is small. However, they concluded that the amount of energy transferred to the bed is negligible in comparison to those areas where a sediment particle impacts the bed directly; in the long term direct impacts are likely to dominate bed erosion. Therefore, parameterizing these open areas is very important to better model bedrock erosion. Our results show that different areas of the bed have different likelihoods of being eroded. Our dataset could be used to develop a probability function that takes into account the effects of local curvature.

    ii)    In the model of Turowski and Hodge (2017), steady state cover is controlled by a characteristic dimensionless mass of sediment, which is equal to the ratio between dimensionless transport capacity and particle speed. This mass is converted to dimensional variables with the help of a characteristic mass, defined by the authors as the minimum mass of sediment required to completely cover the bed per unit area. This minimum mass is likely to be dependent on the ratio between grain size and bedrock macro-roughness.

        Generally speaking, two scenarios are possible: If the grain roughness is larger than the bedrock macro-roughness, the minimum mass of sediment can be determined as proposed by the authors (Turowski and Hodge, 2017; Eq. 34); If the bedrock macro-roughness is larger than the grain roughness, the equation could be adapted by multiplying it by, e.g. $D_{84}/L_{mr}$, to account for the fact that more grains are needed to fill the holes in the bedrock surface. The ratio suggested is based on the value obtained for the sigmoid function relating areal cover and alluvial thickness in the bedrock slab (Eq. 10). The specific grain size chosen and the definition of the macro-

roughness length are issues that need further investigation. The latter issue in particular is still unresolved in the bedrock river literature, where some authors characterize macro-roughness as the standard deviation of the bed elevation signal (e.g. Hodge and Hoey, 2016a, b), but others, such as us, use a characteristic length based on the bed hypsometry (e.g. Zhang et al. 2018, 2015).

## 5 Conclusions

The results of this study lead to the following conclusions:

1. The percent of areal alluvial cover ($p_c$) initially grows rapidly with increasing sediment supply ratio ($q_{st}/q_{bt}$) in meandering channels. Rapid initial growth is likely due to the formation of point bars. Following the formation of these initial deposits, addition of more sediment into the system first promotes the growth of alluvial thickness, and later promotes the growth of the areal extent of alluvial cover. Therefore, a logarithmic relation between these variables reflects their relation better than a linear one. A logarithmic relation allows for rapid initial growth of $p_c$ with increasing sediment supply ratio, but as the sediment supply ratio increases, growth in $p_c$ slows down.

2. The percent areal alluvial cover ($p_c$) as a function of the ratio between alluvial thickness and bedrock macro-roughness ($\eta_a/L_{mr}$) follows an S-shaped (sigmoid) curve. A logistic curve is recommended for models of bedrock erosion that use this framework.

3. The steepness and intersection parameters needed in the logistic curve are likely functions of a characteristic grain size of the alluvium and the bedrock macro-roughness. In this study, the steepness and intersection values used were given by $k \sim L_{mr}/D_{16}$ and $x_o \sim 2.1 D_{84}/L_{mr}$ respectively.

4. Mixed bedrock-alluvial meandering channels may have areas with persistent and transient alluvial cover as well as areas of persistently exposed bedrock. Erosion by abrasion is possible only in the areas with transient alluvial cover. Local normalized $p_c$ values are smaller than reach-averaged values at regions with high curvature and higher at regions with lower curvature.

5. The size and location of the areas of preferential erosion in mixed bedrock-alluvial meandering rivers are a function of sediment supply ratio and local curvature. Low sediment supply ratios are associated with regions of potential erosion located closer to the inner bank. This region moves toward the outer bank as sediment supply increases. High local curvature values are associated with narrow regions of potential erosion whereas lower curvature values are associated with wider regions of potential erosion.

6. The use of either spatially or temporally averaged values of $p_c$, or a combination of both, is not necessarily an appropriate approach to model bedrock erosion by abrasion of bedload. The largest spatial window recommended should be as small as possible so as to capture the local spatiotemporal fluctuations in alluvial cover. The longest temporal window recommended should be quasi-instantaneous so as to capture the temporal fluctuations in alluvial cover.

### 5.1 Future research directions

Based on the results of this study, the following two research directions are proposed:

1. Conduct experiments with the objective of determining appropriate relations to define the steepness and intersection of the sigmoid function for use in numerical models of bedrock erosion based on a framework using ratio of alluvial thickness to bedrock macro-roughness.

2. Develop a model of bedrock erosion by abrasion based on the fluctuations of areal alluvial cover. The model must take into consideration the role of freely-migrating bars and their celerity. The numerical formulation of Inoue et al. (2017, 2016) offers an important advance in this regard.

### Code Availability

The Matlab routines developed to process the time lapse images and eTape data are available at https://doi.org/10.13012/B2-3044828_V1.

### Data availability

All data used in preparation of this manuscript is available at https://doi.org/10.13012/B2-3044828_V1.

### Video supplement

Videos showing the evolution of the bed and the erosion fronts are available at:
https://av.tib.eu/series/606/experiments+on+patterns+of+alluvial+cover+and+bedrock+erosion+in+a+meandering+channel

### Author contributions

Experiments were designed by all authors. R. Fernández conducted the experiments, data analysis and post-processing. Initial manuscript was prepared by R. Fernandez and G. Parker. All authors worked on the final version submitted.

### Competing interests

The authors declare that they have no conflict of interest.

**Acknowledgements**

We would like to thank the Associate Editor Eric Lajeunesse for his patience and feedback during the open discussion and review process. We would also like to thank Jens Turowski and Christian Braudrick for their constructive reviews and very valuable feedback. Participation of all authors in this study was possible thanks to funding provided by the US National Science Foundation [grant EAR1124482]. The authors would like to thank Alejandro Vitale, PhD for his help building the eTape system, assistance with Arduino code development, and preparation of the wiring diagram.

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

**Table 1 Hydraulic parameters common to all experimental conditions**

| Parameter | | Value |
|---|---|---|
| Flow discharge | Q [m$^3$/s] | 0.0123 |
| Channel width | B [m] | 0.60 |
| Centerline depth | H [m] | 0.11 |
| Reach-averaged velocity | U [m/s] | 0.19 |
| Hydraulic radius | $R_h$ [m] | 0.08 |
| Froude number | $F_r$ [-] | 0.18 |

**Table 2 Experiment parameters specific to each run.**

| Run ID | Reach-averaged fraction of cover | Water Temperature | Average bed load transport rate | Water surface slope, middle bend | Water surface slope, entire flume | Kinematic viscosity | Reynolds Number |
|---|---|---|---|---|---|---|---|
| [-] | $p_c$ [-] | $T$ [°C] | $q_{bs}$ [g/s] | $S$ [mm/m] | $S$ [mm/m] | [mm$^2$/s] | $R_e$ [-] |
| pc00 | 0.00 | 24 | 0.00 | 0.99 | 0.68 | 0.9131 | 16,328 |
| pc19 | 0.19 | 20 | 0.08 | 0.97 | 0.79 | 1.0034 | 14,859 |
| pc27 | 0.27 | 24 | 0.25 | 0.99 | 0.75 | 0.9131 | 16,328 |
| pc38 | 0.38 | 27 | 0.55 | 1.14 | 0.88 | 0.8539 | 17,460 |
| pc46 | 0.46 | 21 | 1.47 | 1.19 | 1.01 | 0.9795 | 15,221 |
| pc54 | 0.54 | 22 | - [a] | 1.0[b] | 0.79 | 0.9565 | 15,587 |
| pc72 | 0.72 | 27 | 4.50 | 1.3[b] | 0.97 | 0.8539 | 17,460 |
| pc79 | 0.79 | 24 | 5.60 | 1.3[b] | 0.97 | 0.9131 | 16,328 |

[a] Bed load transport rate not measured for this condition.

[b] Slopes estimated based on the average ratio between middle bend slopes to flume slopes of previous five experimental conditions.

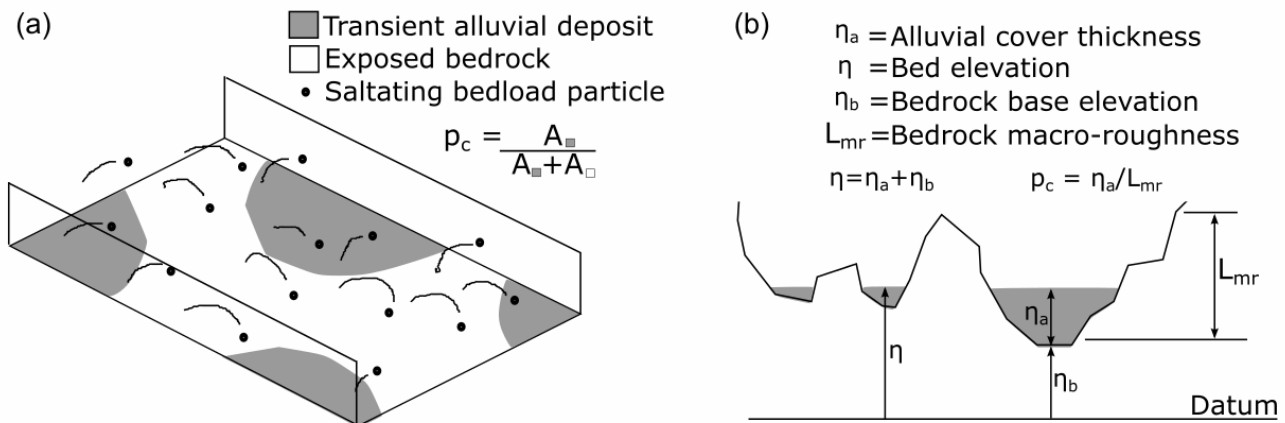

**Figure 1. Schematic representations of (a) the fraction of exposed bedrock showing surface areal cover (Sklar and Dietrich, 2004) and (b) a cross section illustrating filling of a rough bedrock surface with alluvium (Zhang et al., 2015).**

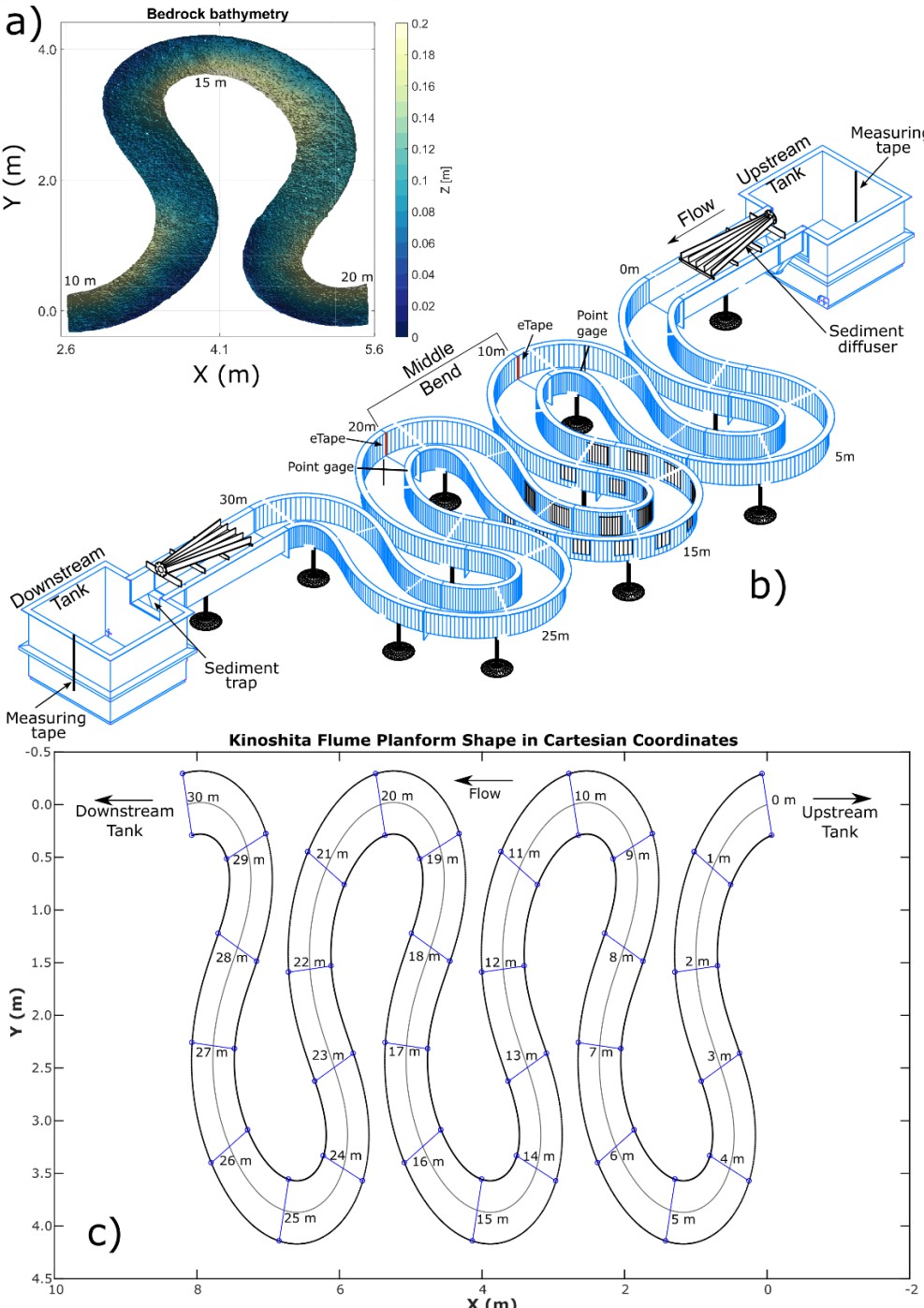

**Figure 2: a) Bedrock bathymetry built inside the Kinoshita flume. Streamwise locations 10m, 15m and 20m are indicated; b) 3D rendering of Kinoshita flume showing location of tank measuring tapes, point gages, eTapes, sediment trap and sediment diffuser, flow direction, and middle bend where all measurements were made; c) Kinoshita shape with streamwise stations indicated. Flow direction from right to left.**

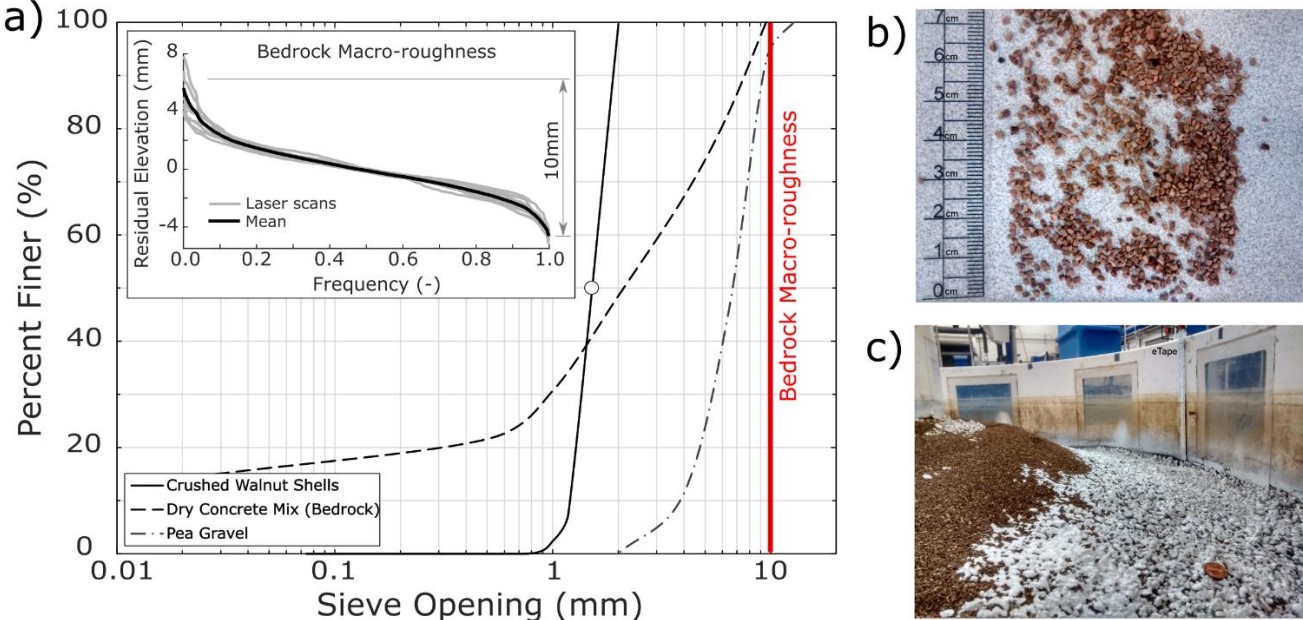

Figure 3. a) Grain size distributions for the alluvium (crushed walnut shells), dry concrete mix used to build the bedrock, and the pea gravel underlying the bedrock basement. Insert shows residual elevations of as-built bedrock bed, measured with laser scans at different cross sections inside the Kinoshita flume. Mean macro-roughness (~10mm) is also indicated in the main plot; b) Image of crushed walnut shells with ruler for scale; and c) Bedrock bed partially covered with alluvium inside the Kinoshita flume.

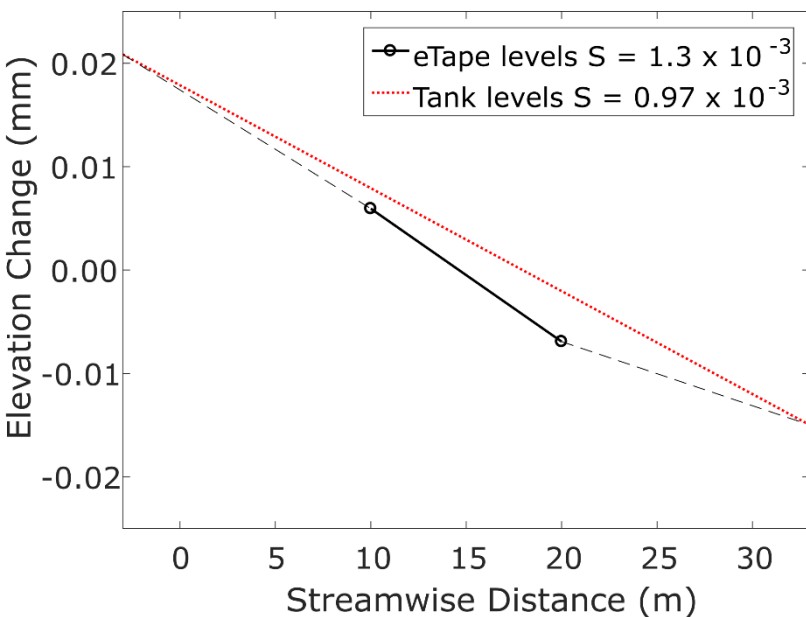

**Figure 4 Average water surface elevation profiles and corresponding slopes based on the eTape readings and the levels measured in the upstream and downstream tanks for Run pc79.**

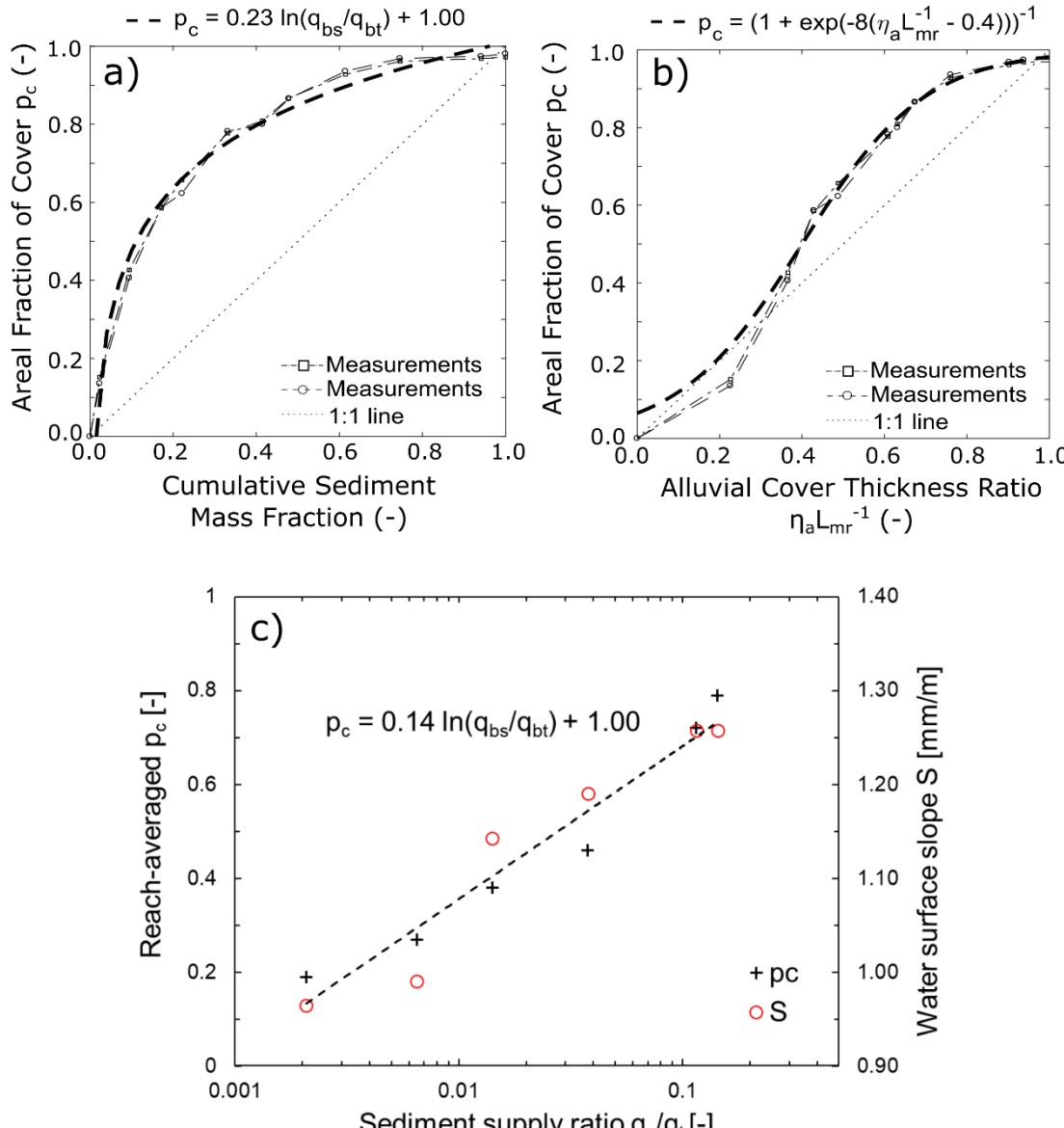

**Figure 5 a) Relation between areal fraction of alluvial cover and cumulative sediment mass fraction for bedrock slab, total mass added to the slab was 646 g and all increments are shown in supplement S1; b) Relation between areal fraction of alluvial cover and the ratio between alluvial thickness and bedrock macro-roughness for bedrock slab; c) Relation between reach-averaged areal fraction of alluvial cover and sediment supply ratio for Kinoshita flume and corresponding water surface slopes as a function of sediment supply ratio.**

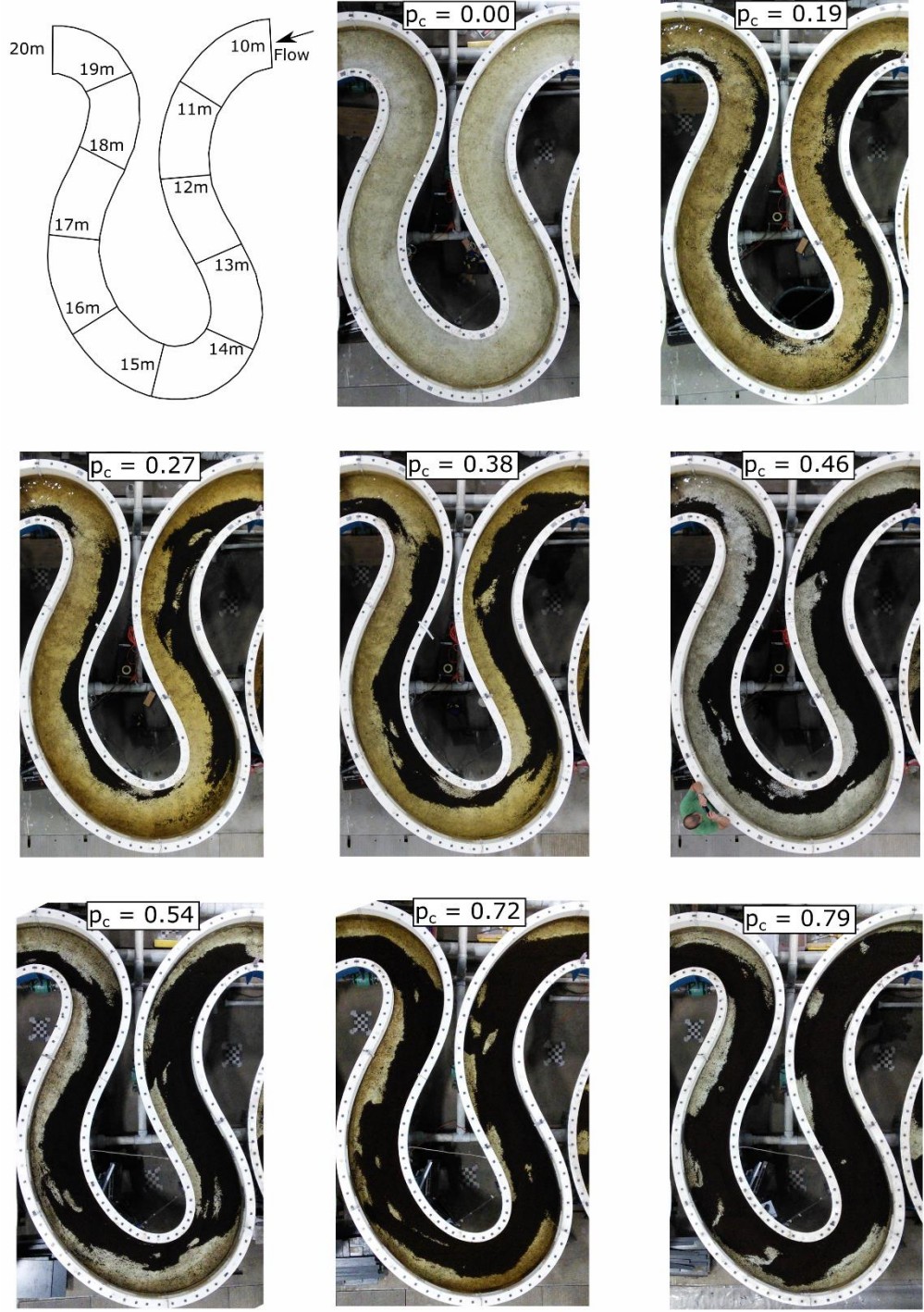

**Figure 6 Images of the middle bend of the Kinoshita flume corresponding to an instant during each of the eight different areal alluvial cover conditions. Volume of sediment in the system grows from top to bottom and left to right. Diagram included at the top-left shows flow direction and contains cross sections indicating the streamwise locations along the middle bend of the flume.**

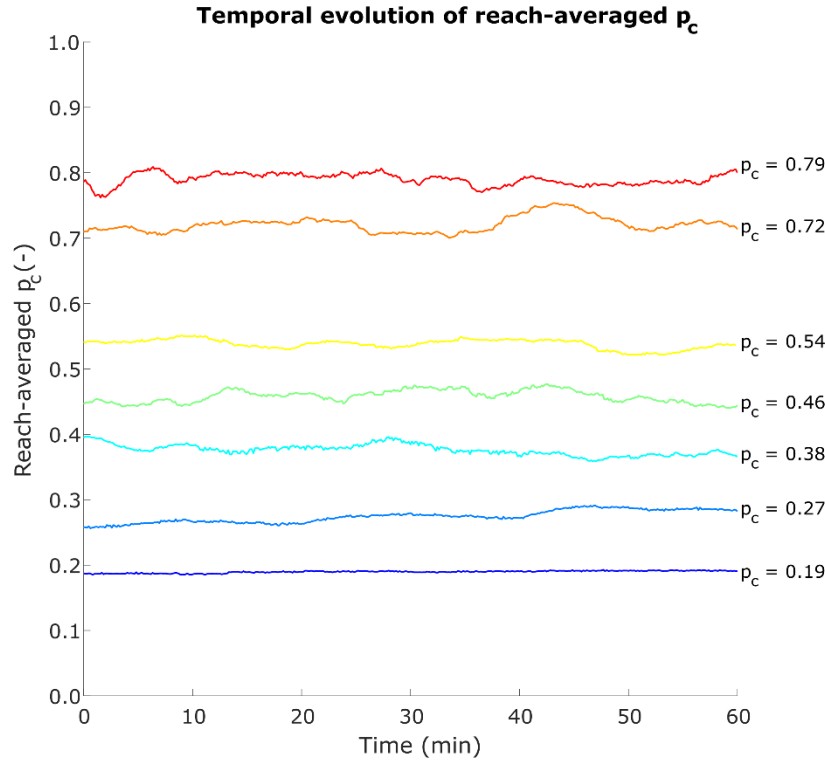

**Figure 7 Temporal evolution of reach-averaged areal fraction of alluvial cover for all experimental conditions that had alluvium.**

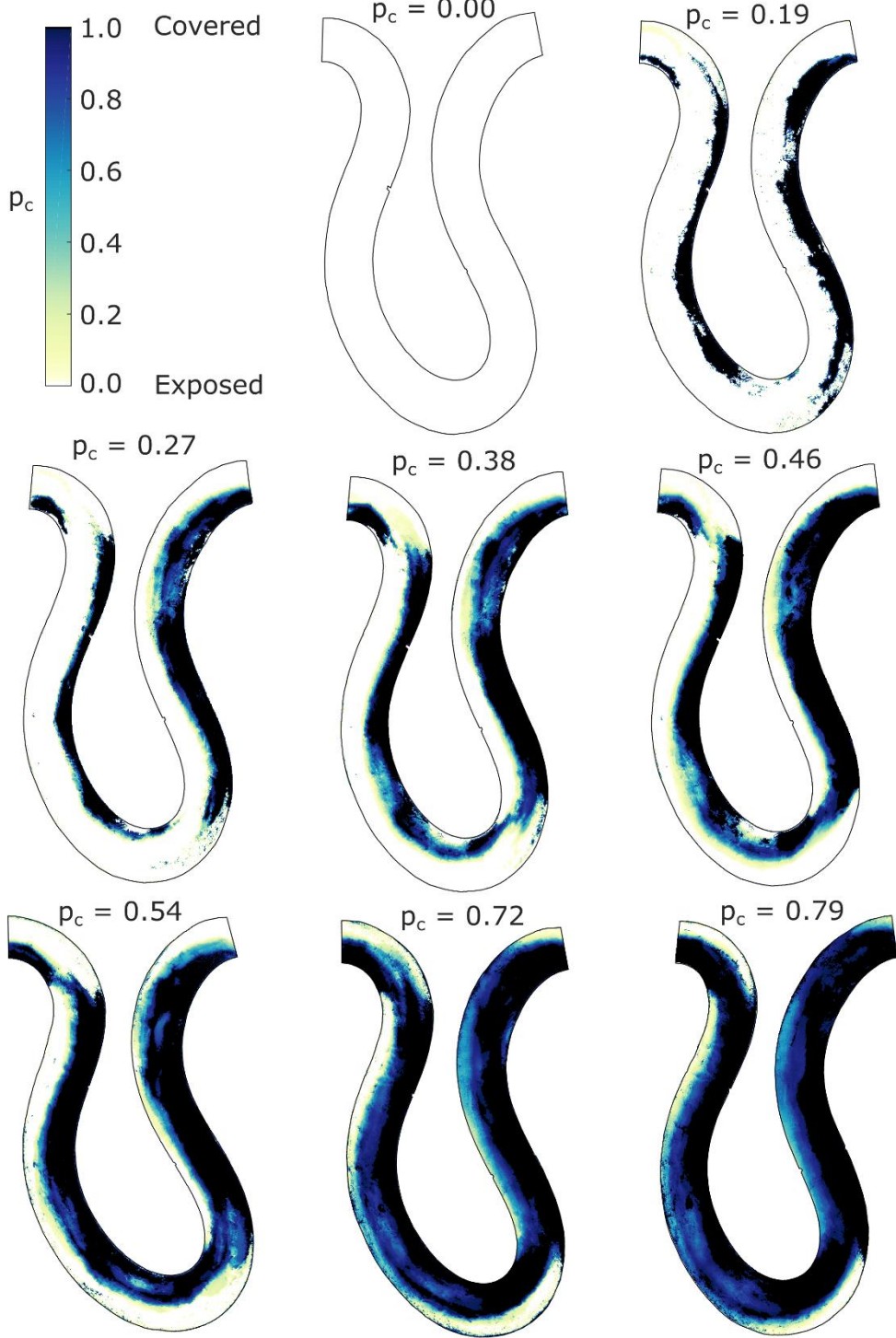

**Figure 8 Maps of spatiotemporal averages of areal fraction of alluvial cover for all experimental conditions.**

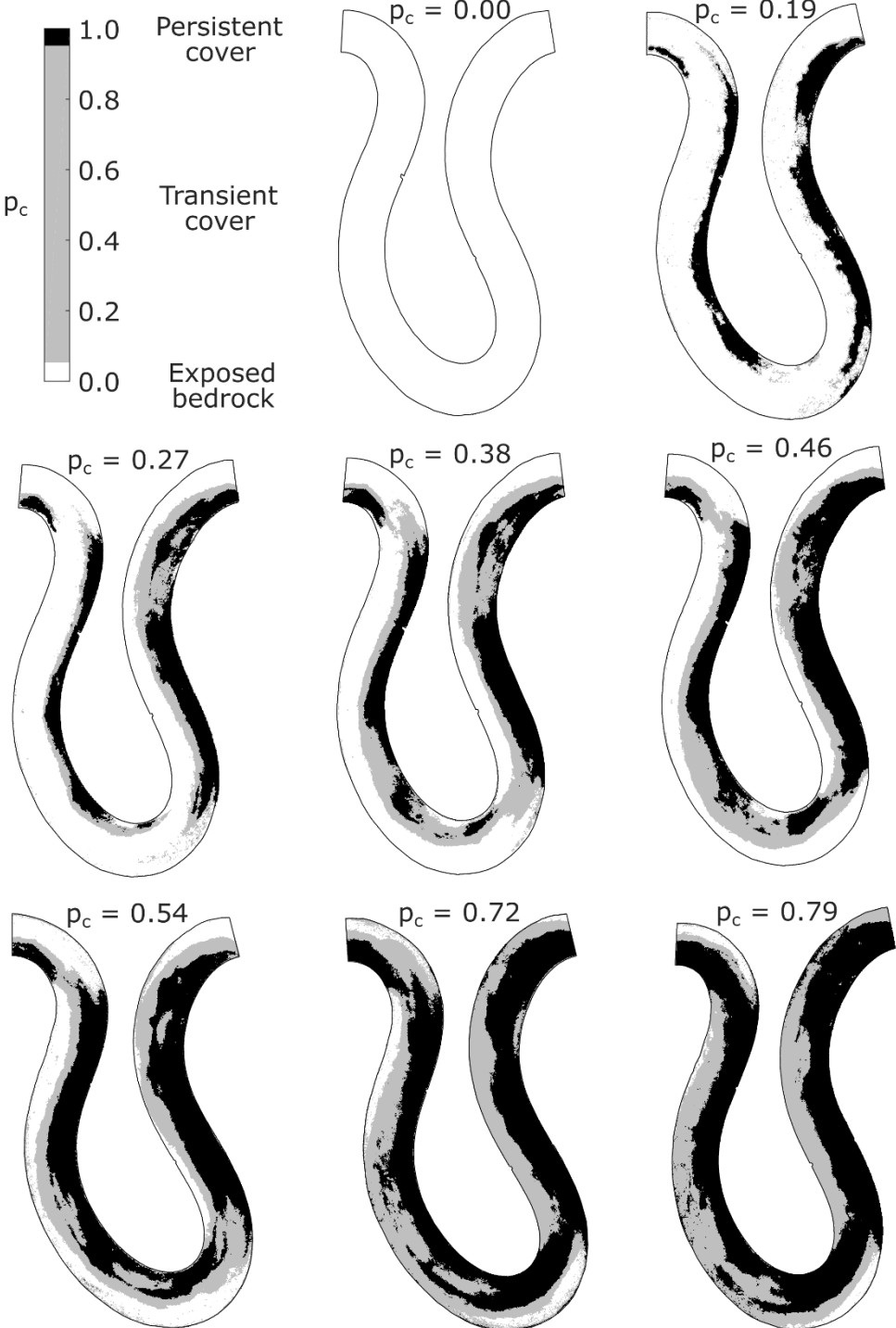

**Figure 9 Maps showing regions with persistent alluvial cover, transient alluvial cover and persistently exposed bedrock for all experimental conditions.**

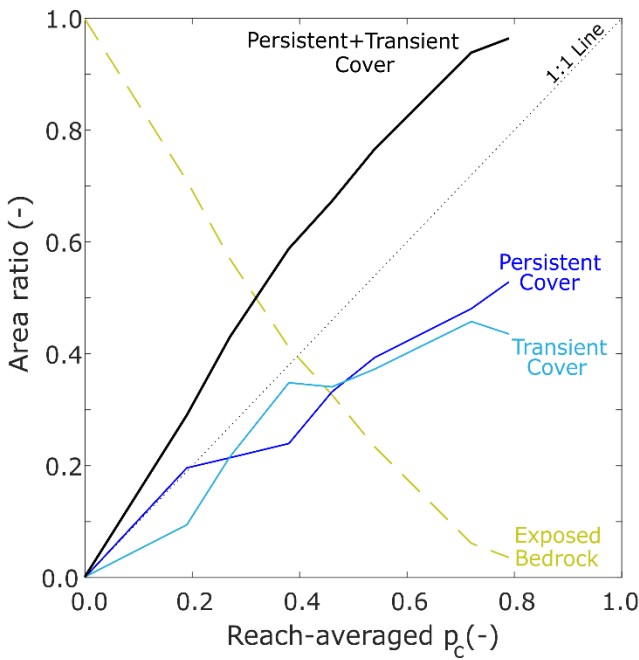

**Figure 10 Reach-averaged area ratios of persistently exposed bedrock, transient alluvial cover, persistent alluvial cover, and persistent + transient alluvial cover as a function of reach-averaged areal cover fraction.**

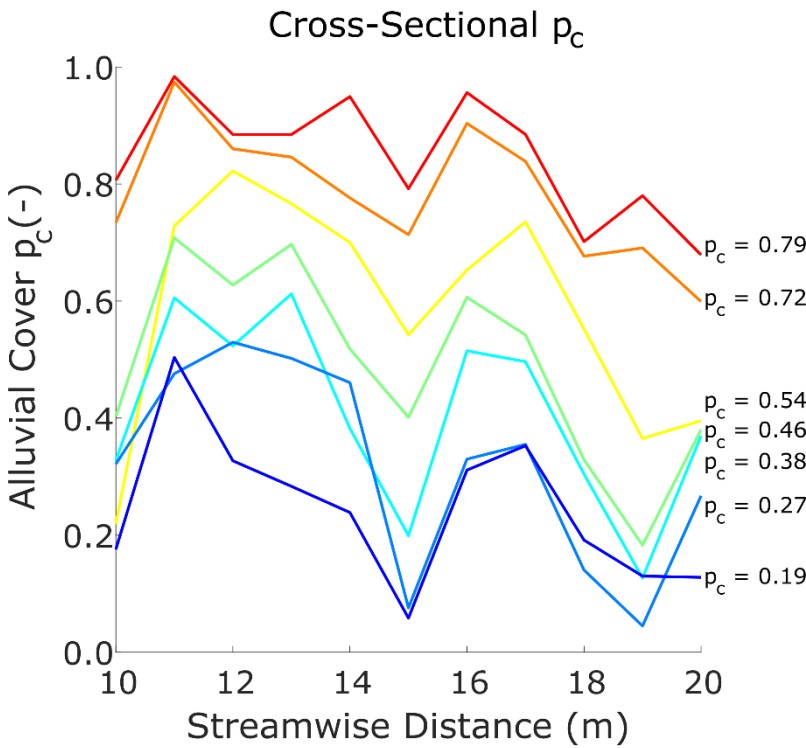

**Figure 11 Cross-sectional averages of areal fraction of alluvial cover for all experimental runs. Local values were extracted every meter between streamwise locations 10 m and 20 m. The legend indicates the corresponding reach-averaged values.**

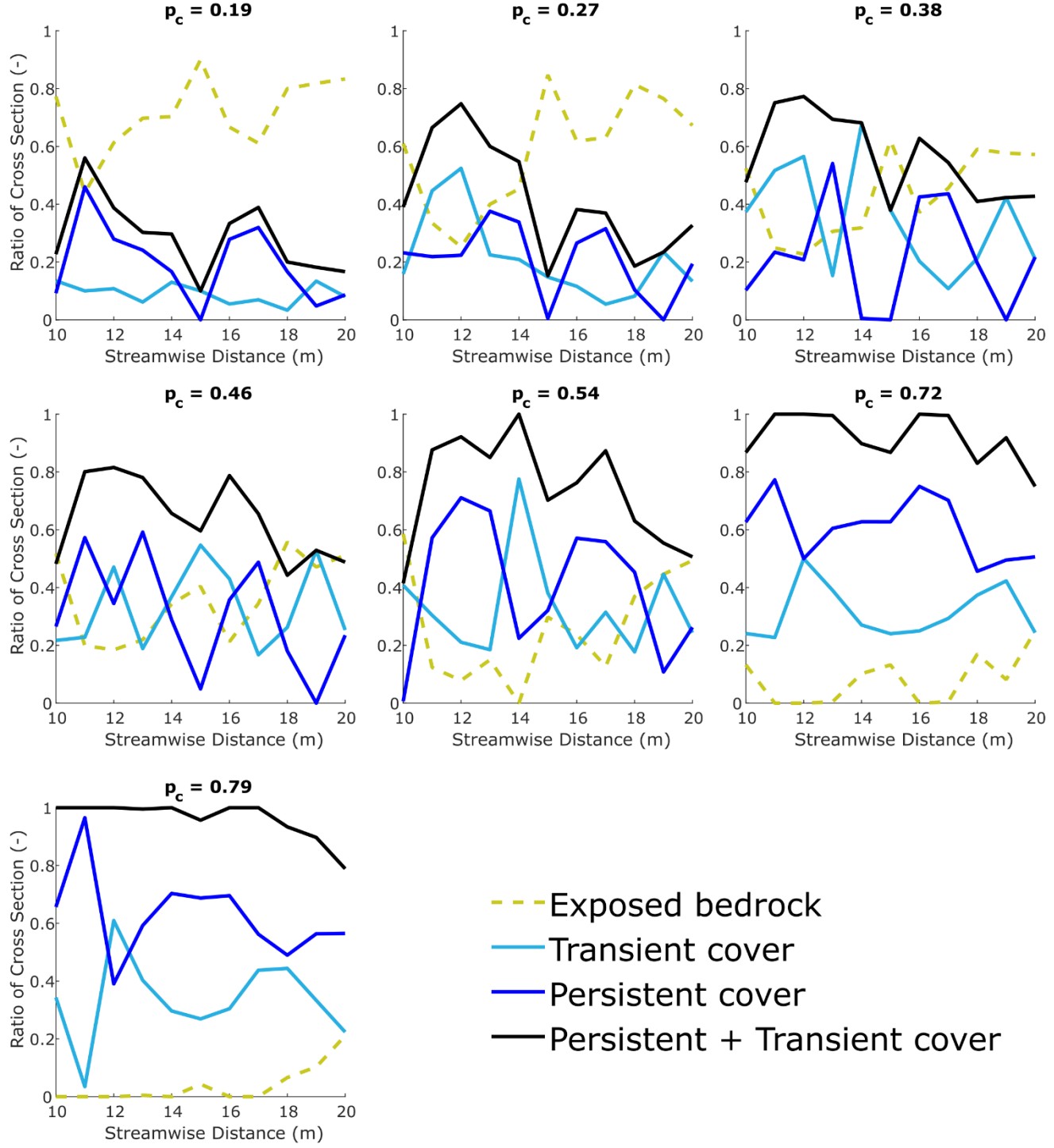

**Figure 12 Cross-sectionally-averaged ratios of persistently exposed bedrock, transient alluvial cover, persistent alluvial cover, and persistent + transient alluvial cover for all experimental conditions.**

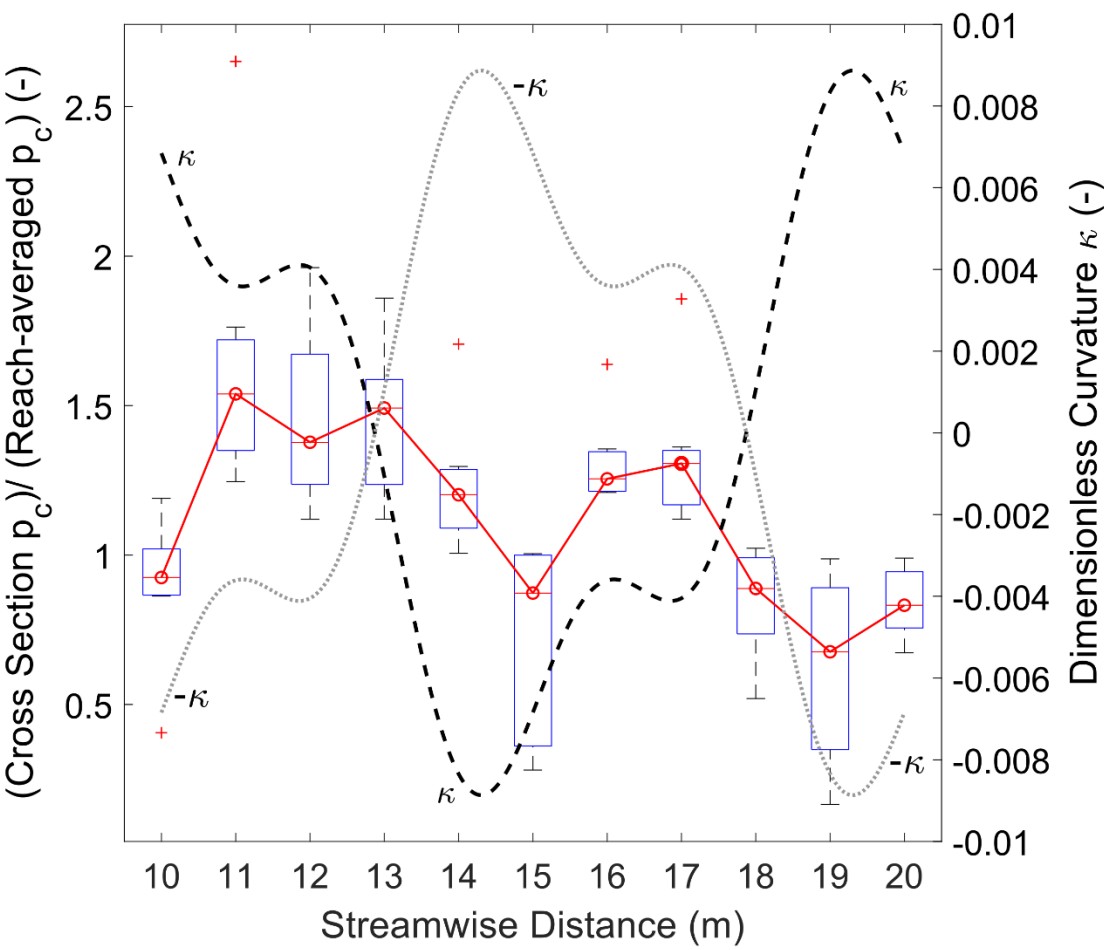

**Figure 13 Boxplots of normalized cross-sectionally-averaged areal fraction of alluvial cover in the middle bend of the Kinoshita flume. The dimensionless curvature of the flume κ and its negative value - κ are plotted to better show the salient trends.**

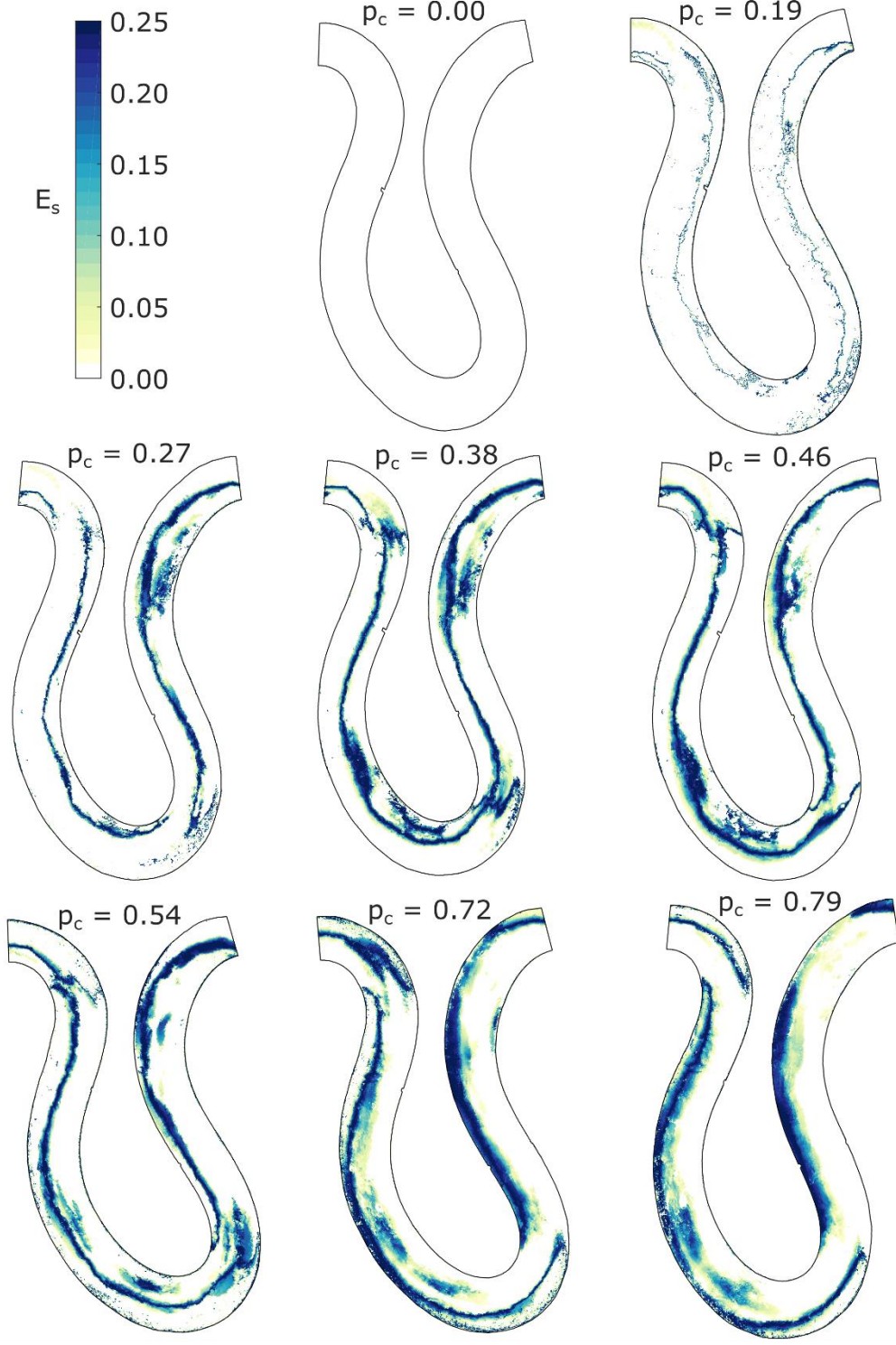

**Figure 14 Maps of spatiotemporally-averaged erosion potential for all experimental conditions.**

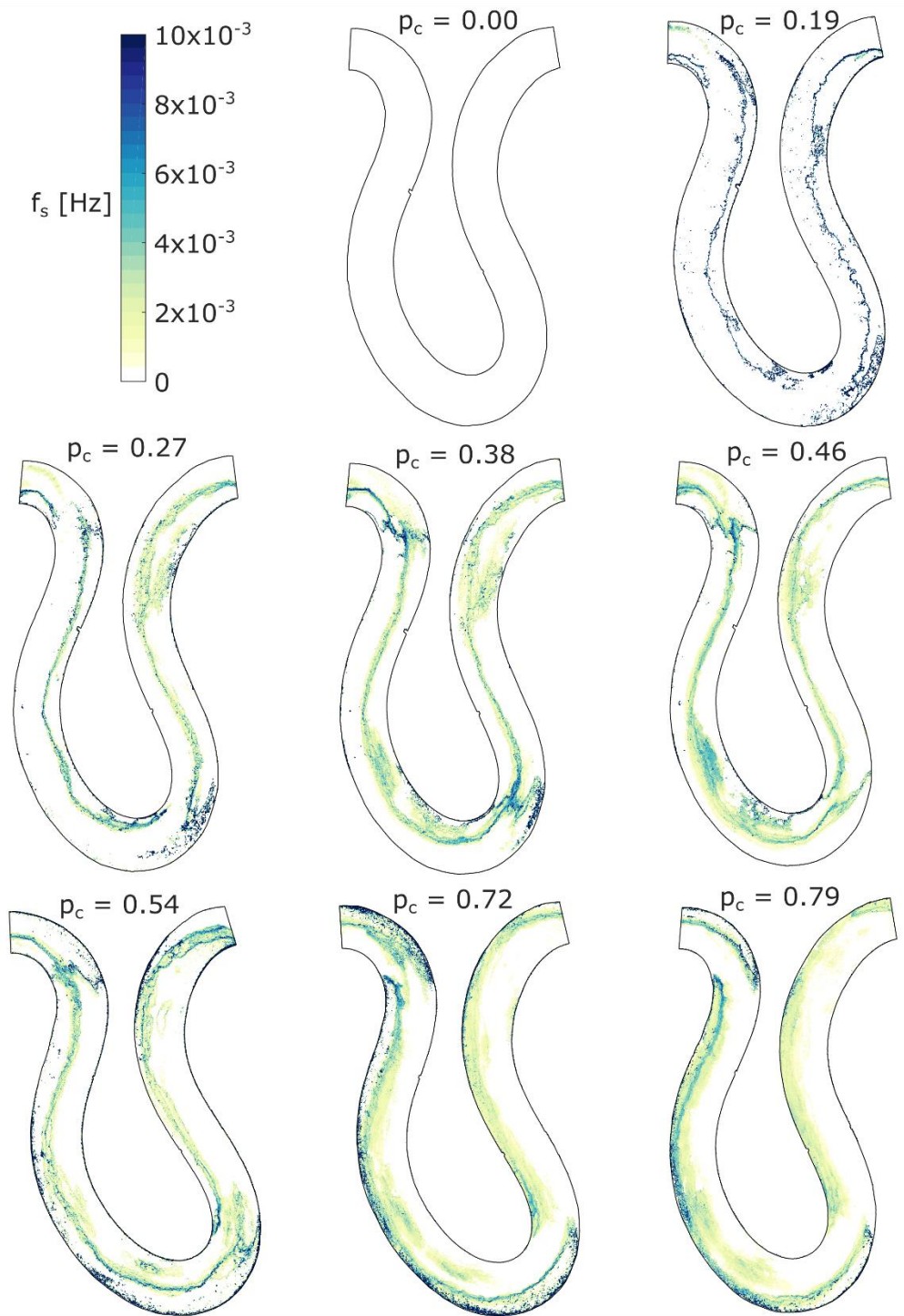

**Figure 15 Maps of frequency of strikes for all experimental conditions. Frequency shown is based on number of images. Dividing the values by 10 s, which is the time between images, will give the actual frequency in Hz.**

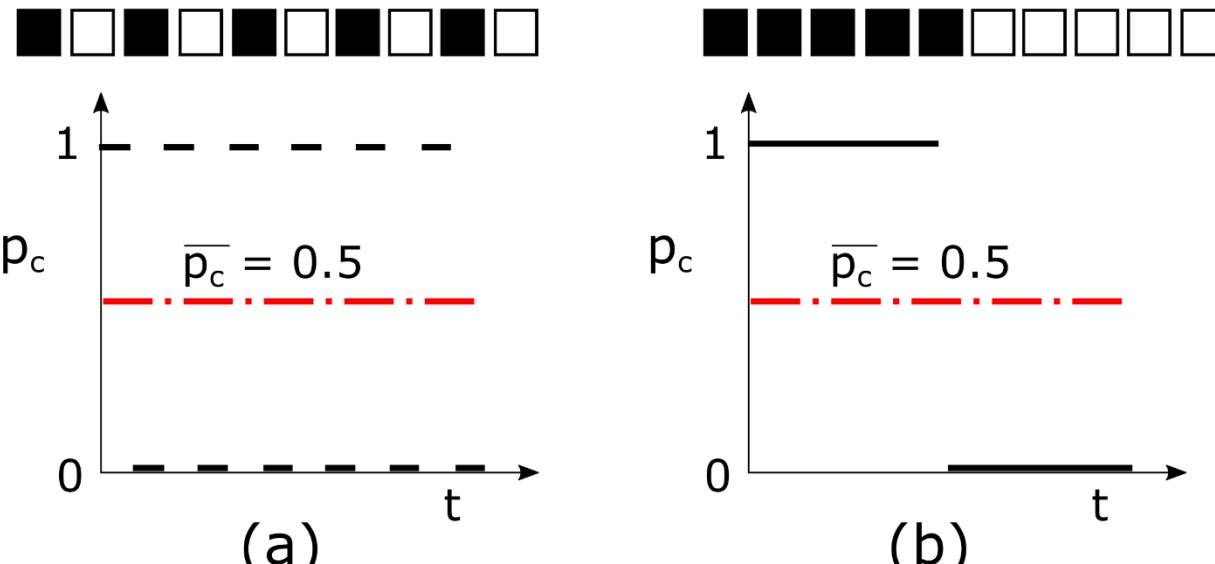

**Figure 16 Simple example showing that temporal averages alone of the areal cover fraction of alluvial cover are insufficient to quantify bedrock incision. Bed conditions (a) and (b) have the same average cover but (a) would experience more erosion than (b) due to a greater frequency of fluctuations in alluvial cover.**

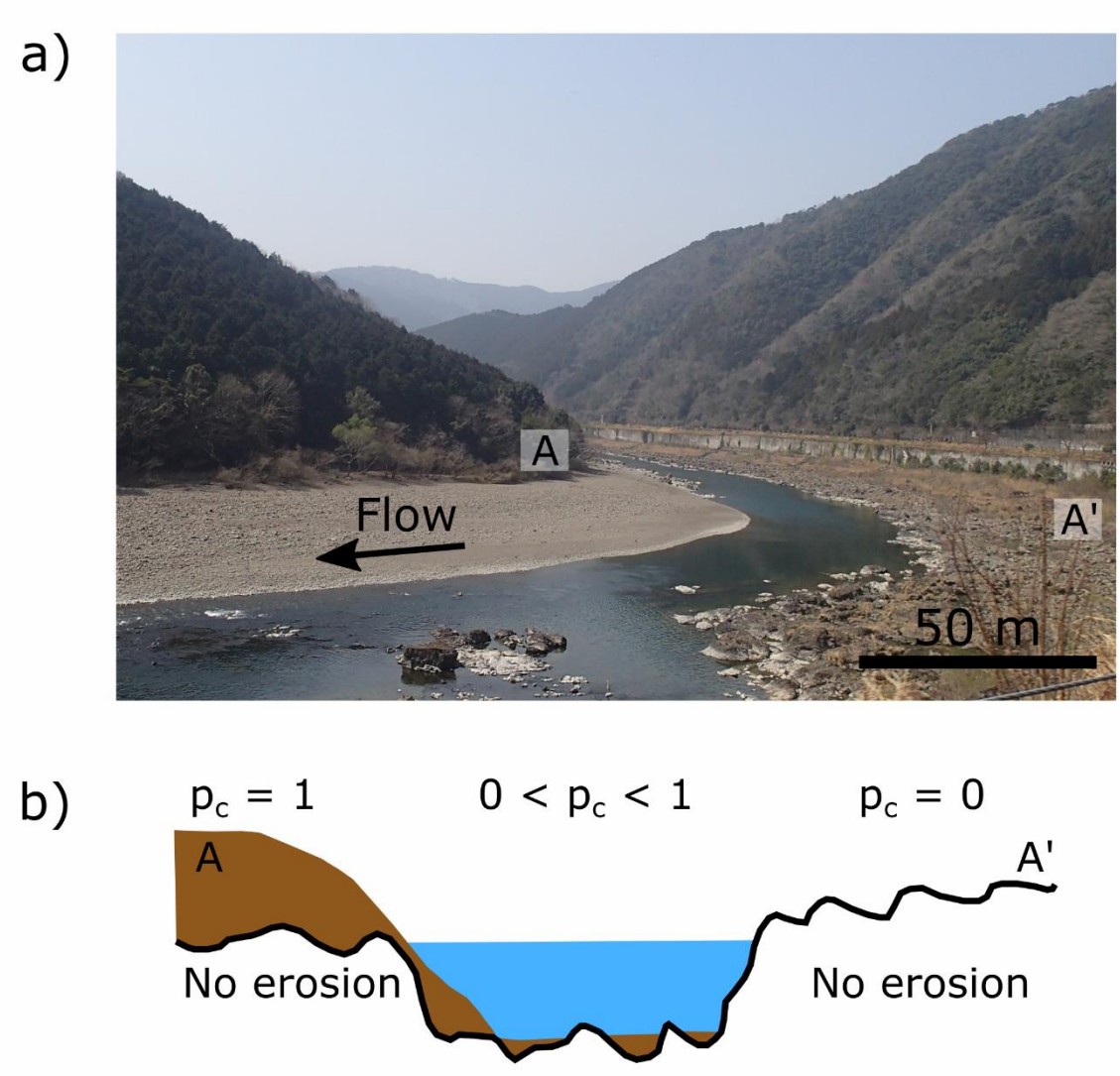

**Figure 17 a)** Image of a reach of the Shimanto River, Shikoku, Japan showing partial cover with alluvium. **b)** Sketch of cross section A-A' (with strong vertical exaggeration) indicating inferred regions of erosion and no erosion.