# Peer review of "Experiments on patterns of alluvial cover and bedrock erosion in a meandering channel"

_Earth Surface Dynamics, 2019_

## Referee Comment (RC1) · Jens Turowski (Referee) · 1 Mar 2019

Fernandez et al. conducted experiments to investigate the spatial and temporal distribution of bed cover in a bedrock meander bend. The study is timely and highly needed, given that there are very few published data on this particular topic. That said, the paper left me wanting for a number of reasons. While results are described in meticulous detail, interpretations, implications, and a placement into the body of current knowledge are largely lacking. While the authors cite a large number of appropriate references in the introduction, I do not have the impression that these were read in connection with the presented analysis. The entire discussion – six pages long – features a lonely five references and actually reads more like a continuation of the results. This is the more surprising as some of the references cited in the introduction actually describe models

and frameworks that are highly relevant for the study. While this is the major issue I have with the paper, there are a few other important points. For example, there is a mixture of methods, results and interpretation in the results and discussion sections, and the method description is thin and unclear in places. As a summary, I think the paper needs some clarifications and re-structuring, and, in particular, a framing of the results in the context of existing literature.

The data that the authors have produced lend themselves for applying and testing recent theoretical work. In this paragraph, I want to highlight some very relevant work. I apologize in advance that this will carry quite a number of references to my own papers; I have spent time thinking about the cover effect for my entire career and have written 8 first-author papers on it (and yes, I tried to stop a few years ago, but it did not work...). Please don't feel obliged to cite everything mentioned below in your paper. Have a look and work with what you think is relevant.

1) There is some work upscaling cover from the grain scale using models. This is highly relevant to your work (and the lack of it is even used in the motivation of the study). I know of three studies: Turowski (WRR, 2009), Turowski and Bloem (Geodin. Acta, 2016) and Hodge and Hoey (JGR 2012).

2) Recent work from the group of Hodge and Ferguson at Durham that focusses on field data may also be of interest (Hodge and Hoey, 2016; Hodge et al., 2011, 2016; Ferguson et al., 2017).

3) Rebecca Hodge and I (Turowski and Hodge, ESurf 2017) have recently developed a theoretical framework that seems to be ideal to assess the authors' data set. This framework completely separates the issue of cover from sediment transport dynamics. We also defined benchmark cases and describe physical reasons of why a particular cover curve may deviate from these. Your results and the interpretations you propose seem to align with this. And yes, I am obviously biased, but let me know what you think about it.

[Figure]

4) There is a great little experimental paper on the interaction of erosion, channel morphology and cover in a meandering channel by Shepherd (Science, 1972; see also the follow up by Shepherd and Schumm, GSAB 1974). This is often overlooked but highly relevant.

Before I make a few more detailed comments specific to the manuscript, I would like to address some common misconceptions about the cover effect that persist in the literature and also shine through here.

1) There is a need to distinguish the relationship between cover and sediment mass on the bed from the relationship of cover with the ratio of sediment supply to transport capacity. Although we claimed in a previous paper (Turowski et al., JGR 2007) that these are equivalent, this is not true. Turowski and Hodge (ESurf 2017) derived a formally correct transformation between these two functions. This point affects the discussion, for example the comparison with the linear cover model (that is typically formulated as a function of the ratio of sediment supply to transport capacity).

2) The exponential cover relation arises when assuming that sediment is randomly distributed on the bed, i.e., each part of the bed has the same probability of deposition. This is true for the relationship between cover and sediment mass, NOT for the relationship of cover with the ratio of sediment supply to transport capacity. The exponential function arises whether one considers the static or the dynamic cover effect, or both combined. There is no relation between the exponential cover relation and the assumption of a dynamic cover effect.

3) The exponential function was derived under the assumptions that flow, sediment supply, bed topography (slope, roughness...) can be considered constant within the area of interest. It is clear that spatial variability should change the functional relationship.

4) We put forward the dynamic cover effect using the common framework of a subcapacity flow (detachment-limited assumption). In this case, capacity is larger than

supply. The river has spare energy for transport and uses this to entrain any stationary sediment. This is the case when assuming that in the control area that is considered, all relevant parameters (hydraulics, supply...) can be considered to be constant (see point 3). In the strict use of these assumptions, a static cover is not possible. This can be resolved by working, for example, in an entrainment-deposition framework (see Turowski, WRR 2009; Shobe et al., GMD 2017; Turowski and Hodge, ESurf 2017).

Specific comments (page.line)

2.21 This is a misunderstanding. The exponential cover term arises under the assumption that deposition is equally likely on each part of the bed. This is independent of whether there is static or dynamic cover or both.

2.22 Dieter Rickenmann and I presented some field evidence for the dynamic cover effect in a paper in 2009 (Turowski & Rickenmann, ESPL, 2009).

2.22 The model of Lague is equivalent to the linear / exponential area-based models when working with the mean sediment thickness.

2.25 ff. This paragraph reads like a random selection of statements. The relevance and the main flow of argument are unclear.

2.30 ...presented...

3.9 Shepherd, Science 1972, did some great experiments looking at cover and erosion in a meandering channel. Sadly, his work is often overlooked. There is follow-up paper by Shepherd and Schumm (GSAB 1974).

3.12 The issue of scale is ok and I agree that this has been insufficiently addressed. Nevertheless, Hodge and Hoey (JGR 2012) and I (Turowski, WRR 2009; Turowski and Bloem, Geodin. Acta 2016) used models to look at cover dynamics from a grain perspective. The assumption in this work is that the appropriate area or length scale to average cover over is much larger than the grain size. It would be easy to assess how much larger it needs with the models developed in the mentioned papers.

3.14 It is my impression that here two separate issues are confounded. The term 'transient' is ambiguous, because it could mean that a particular patch of cover is created and destroyed over time, or that the sediment in the patch is continuously replaced by new sediment. The relevant point here is not that cover is transient – most model deploy a temporal average or a stochastic notion of a steady state – but that cover patches in a bend may persist in location and thus time-averaged incision varies along and across the channel. This means that, over the long term, the bed is not lowering at the same rate everywhere.

3.16 I do not have the impression that this is not acknowledged by the modelers.

3.18 This is a good question, and again, there is some work setting up the basics for this. The above-mentioned stochastic models of resolve these aspects explicitly and in a previous paper (Turowski, WRR 2009), I derived probability distributions of cover and erosion rates. Temporal fluctuations in cover are also modelled by Lague (JGR 2010). The model of Turowski and Hodge (ESurf, 2017) also predicts the temporal evolution of cover. In the latter paper, a mass balance framework is also presented that can easily be extended to look at cover dynamics along and across a channel.

4.3 This notion (transport can happen only on covered portions of the bed) is only true if a dynamic cover effect is assumed. Even then, I find the formulation highly misleading. Clearly, sediment can transported over a clean bedrock bed (think about a single particle driven by water in an otherwise empty flume!), and the notion would imply that there always needs to be a long-stream connected cover. There are many observations suggesting that this is not the case.

4.21 I suggest a more careful formulation here. Yes, in channel bends, there can be stationary deposits. For a given geometry, the location and size of these is dependent on discharge (or the ratio of discharge or capacity to supply). If the term 'mixed bedrock-alluvial channels' indicates a bedrock channel with a partially covered bed, we would need to integrate over the variable discharge to calculate long-term incision

rates. During the highest floods, the entire channel may still be stripped of cover.

5.10 Information on methods and set up is incomplete. What was the size of the sediment (median, distribution)? Was the foam erodible? Why foam? Which type of foam? How were the foam sections linked (the description suggests that there where sharp steps every 0.5m)? What was the initial pattern of gravel in the flume (spatial distribution, thickness)? Why was this pattern chosen (does the initial distribution affect the results)? Did the concrete completely cover the gravel or did bits stick out, contributing to roughness? Can you give some more information on the concrete that was used (e.g., the sand fraction may affect the bed roughness)?

5.12 How were bedforms averaged out? How were bedforms defined?

5.16 What is the 'maximum of the foam'?

5.23 Please give information on the model and a reference.

5.23 Can you be precise in giving the precision?

5.23 Why and how were these locations chosen? Are these line (cross-sections?) or areal scans?

5.24 I do not understand the exact protocol here. Why a 4th order polynomial?

5.25 Why do the authors think that this particular protocol gives and adequate measure of the macro-roughness?

6.18 If you know the weight, why don't you tell the reader? There is no need to be secretive.

6.18 I have trouble understanding the rationale of this experiment. Please explain more clearly what you intended to do. From the writing, it seems that you wanted to measure the minimum mass of sediment necessary to fully cover the bed, correct? I guess I was confused by the term 'sediment supply ratio'; my first impression was that you wanted to measure sediment supply divided by transport capacity (which would be the

typical definition of the term 'sediment supply ratio'). It be would be good to explain the intention of the experiment clearly, including the physical or theoretical reasoning behind it (see Turowski and Hodge, ESurf 2017, for a discussion of why the minimum mass necessary to cover the bed is a useful normalization factor).

6.22 How was macro-roughness measured?

6.25 Why did you think that a single repetition was sufficient to mitigate for human errors?

8.10 and following. I suggest to use 'persistent' instead of 'permanent' throughout the paper. The cover was not permanent, because it was established throughout the run and removed after it. The dictionary gives a definition of permanent as "lasting or intended to last or remain unchanged indefinitely".

8.18 Are these percentages applied to spatial or temporal variance?

8.18 This information should go into the method section.

8.22 The legend of the figure should go into the caption. Here, better describe the main features that you want to highlight for the reader.

9.12 Section 3.5 does not give results. The way of calculating erosion potential should be moved to the method section.

9.13 Again, the legend of the figure should be placed in the caption. In the main text, describe the results that you want the reader to see on the figure (e.g., 'A correlates with B', or 'A shows a maximum at a value of B = XX').

9.24 dot missing after 'al'

10.6 The logistic curve is defined over the space from minus to plus infinity, while the cover is defined between 0 and infinity. Further, we know that (in Fig. 5b), the curve should go through the point 0,0. In a recent paper, Rebecca Hodge and I developed a mathematical framework to deal with such cases (see Turowski and Hodge, ESurf,

2017).

11.24 and following. Yes, and this is not really surprising. What are the implications, for example for meander development and channel width?

References

Ferguson, R. I., Sharma, B. P., Hodge, R. A., Hardy, R. J., and Warburton, J.: Bed load tracer mobility in a mixed bedrock/alluvial channel, J. Geophys. Res.-Earth, 122, 807–822, https://doi.org/10.1002/2016JF003946, 2017.

Hodge, R. A. and Hoey, T. B.: Upscaling from grain-scale processes to alluviation in bedrock channels using a cellular automaton model, J. Geophys. Res., 117, F01017, https://doi.org/10.1029/2011JF002145, 2012.

Hodge, R. A. and Hoey, T. B.: A Froude scale model of a bedrock-alluvial channel reach: 2. Sediment cover, J. Geophys. Res., 121, 1597–1618, https://doi.org/10.1002/2015JF003709, 2016.

Hodge, R. A., Hoey, T. B., and Sklar, L. S.: Bed load transport in bedrock rivers: The role of sediment cover in grain entrainment, translation, and deposition, J. Geophys. Res.-Earth, 116, F04028, https://doi.org/10.1029/2011JF002032, 2011.

Hodge, R. A., Hoey, T. B., Maniatis, G., and Leprêtre, E.: Formation and erosion of sediment cover in an experimental bedrock-alluvial channel, Earth Surf. Proc. Land., 41, 1409–1420, https://doi.org/10.1002/esp.3924, 2016.

Shepherd, R. G.: Incised river meanders: Evolution in simulated bedrock, Science, 178, 409–411, https://doi.org/10.1126/science.178.4059.409, 1972.

Shepherd, R. G., Schumm, S. A.: Experimental study of river incision, Geol. Soc. Am. Bull., 85, 257-268, 1974.

Shobe, C. M., Tucker, G. E., and Barnhart, K. R.: The SPACE 1.0 model: a Landlab component for 2-D calculation of sediment transport, bedrock erosion, and landscape

evolution, Geosci. Model Dev., 10, 4577–4604, https://doi.org/10.5194/gmd-10-4577-2017, 2017.

Turowski, J. M.: Stochastic modeling of the cover effect and bedrock erosion, Water Resour. Res., 45, W03422, https://doi.org/10.1029/2008WR007262, 2009.

Turowski, J. M. and Bloem, J.-P.: The influence of sediment thickness on energy delivery to the bed by bedload impacts, Geodin. Acta, 28, 199–208, https://doi.org/10.1080/09853111.2015.1047195, 2016.

Turowski, J. M. and Hodge, R.: A probabilistic framework for the cover effect in bedrock erosion, Earth Surf. Dynam., 5, 311–330, https://doi.org/10.5194/esurf-5-311-2017, 2017.

Turowski, J. M., Lague, D., and Hovius, N.: Cover effect in bedrock abrasion: A new derivation and its implication for the modeling of bedrock channel morphology, J. Geophys. Res.-Earth, 112, F04006, https://doi.org/10.1029/2006JF000697, 2007.

---

## Referee Comment (RC2) · Jens Turowski (Referee) · 11 Mar 2019

Dear authors,

I just discovered this report, which may be relevant for your paper: https://www.ethz.ch/content/dam/ethz/special-interest/baug/vaw/vaw-dam/documents/das-institut/mitteilungen/2010-2019/245.pdf

The author did experiments with changing sediment supply in a meandering channel (smaller sinuosity than yours), investigating deposition patterns over a flat bed and the subsequently developed armoured bed, which was artificially fixed in some of the experiments. The relevant chapter is chapter 7.

All the best, Jens

---

## Referee Comment (RC3) · Christian Braudrick (Referee) · 12 Apr 2019

**Christian Braudrick (Referee)**

cbraudrick@gmail.com

Received and published: 12 April 2019

The paper "Experiments on patterns of alluvial cover and bedrock erosion in a meandering channel" by Fernández et al. to explore the potential erosion of bedrock by moving sediment in a sinuous flume. As the sediment was varied, photographs were used to track areas that were (1) permanently covered in sediment and therefore unlikely to erode (2) sediment was continually absent and lacked the tools to erode bedrock, and (3) areas that had transient cover and were therefore subject to erosion. The figures are really fantastic and very clearly show the results. The main issue with the paper is the structure, and I think this issue is easily remedied. The current results section describes the axes and plots of the figures but does not describe the results, as these descriptions are included in the discussion. I found this approach confusing and

think the figure description from the discussion can be moved to the results, and then the paper will be a significant contribution to our understanding of bedrock erosion in sinuous rivers. A very interesting result that was not discussed much in the paper is that the zone of transient cover moved toward the outer bank when sediment supply and curvature were high enough. This is a cool result and a discussion of this observation would be a valuable asset to the literature. The paper also contains a second experiment where sediment was added to a flat slab to develop a relationship between sediment mass added and cover, and the areal fraction of cover and the thickness relative to the roughness of the bedrock. These results are interesting and should be very useful for future model development, but the motivation for these experiments could be laid out more clearly at the outset, it wasn't clear to me until the results section. In addition, more overtly labeling the figures representing the slab experiments would help. A picture of the slab would also be useful. I am jealous of the experimental design and thought the paper was interesting and certainly merits publishing. A few of the methods and materials need more explanation to properly assess the scaling of the experiments. Specific questions and comments are listed below. Specific questions/comments I had several questions about hydraulics and scaling which would be helpful: 1. What is the settling velocity of the particles (calculated or measured). How does that influence the observations? 2. The Shields stress is relatively high ( $\sim 0.18$ ) does that have an impact on the hop length of the particles, and the locations where impacts are expected? Does the density of the particles affect the hop length? 3. The width depth ratio is relatively narrow ( $\sim$ 5.5), yet migrating bars are described in the paper. IS there any data on the width depth ratio of bedrock meanders? Are the bars forced bars or free bars (or some other bedform)?, and would the results in a wider channel differ? 4. I would be interested to hear more description of the channel morphology including how deep were the pools and how much did the lateral slope of the bar vary. To some degree, experiments are their own self-contained system, but I couldn't help asking myself how the results would differ if sand or gravel were used rather than walnut shells. Presumably the area of transient cover might be closer to

**ESurfD**
the inner bank with heavier sediment? Questions about experimental procedure: 1. Approximately how thick was the pea gravel? 2. Was the bed cleaned out between runs? Is the adjustment between runs reflected in any of the results (i.e., how long did it take to adjust the channel morphology)? 3. My memory of walnut shells is that they are pretty angular. Does that affect the experiments at all?

Other specific comments Page 5. Lines 10-21. I found the intermingling of "bed material", "alluvium", "bedrock", and "bedrock basement" confusing. Bed material could either refer to the bedrock or alluvium. I think sticking to bedrock or alluvium would be helpful. I also got confused by the way the description of the artificial bedrock was built starting with the bottom. It might be helpful to start by describing what each component is used for (walnut shells to give the basic channel shape (pools and bars), pea gravel to provide roughness, and concrete to provide strength), then describe how it was built. How thick was the pea gravel? Page 5. Line 19. Is there anything special about the cement mix? What was the ratio of water to sediment (could be useful to future experiments). Section 2.3. What was the scanning interval of the bed, and was the scanner measuring a grid or cross sections (from later in the paper I gathered it was cross sections but I am not sure)? Page 7, line 21-23. The slope was measured with point gages from 9 m to 21 m. These spots are just before the apex of one bend and after the apex of another bend. Often the water surface elevation can vary based on position in the bend as water backs up behind the bend apex. Did this occur in these experiments? If so, are these slope measurements representative of the overall slope. Page 10, lines 5-15. Does the thickness of the alluvial cover depend on the shape of the particles? Figure 10. Please define area ratio in the figure legend. Figure 11. Can we quantify the variability of cross sectional alluvial cover relative to reach average cover using standard deviation, or just a range? Figure 13. I found myself curious how symmetrical the results were and wondered why the cover was higher upstream than downstream. How did cover vary as a function of the absolute value of curvature? Presumably the curvature upstream and downstream matters as well as local curvature.

**ESurfD**

**ESurfD**

---

## Author Response (AR1)

25 June, 2019

Dr. Eric Lajeunese

Associate Editor

Earth Surface Dynamics

Dear Eric:

Thank you very much for your patience with us while we revised the manuscript. We have made our best effort to work with the reviewer's suggestions to improve the quality of our manuscript. We have:

1. Restructured the paper following the reviewers' comments and have extended the discussion to link our work to the most relevant work associated with it.
2. Added two new supplemental materials which extend our 'Materials and methods' section and provide more clarity.
3. Responded to the reviewers' comments point by point. Please find below our answers to both Jens and Christian.

We believe this revised version meets the high quality standard of ESurf and look forward to its publication.

Warm regards,

Roberto Fernández

On behalf of all authors

Answers to Jens Turowski (second set):

We have copied Jens' comments into this document and have provided answers/comments using ***bold italics*** fonts. References follow Page/Line (P.L) format unless otherwise specified.

1) There is some work upscaling cover from the grain scale using models. This is highly relevant to your work (and the lack of it is even used in the motivation of the study). I know of three studies: Turowski (WRR, 2009), Turowski and Bloem (Geodin. Acta, 2016) and Hodge and Hoey (JGR 2012).
***We have revisited these papers and added references to them in the discussion. Thank you.***

2) Recent work from the group of Hodge and Ferguson at Durham that focusses on field data may also be of interest (Hodge and Hoey, 2016; Hodge et al., 2011, 2016; Ferguson et al., 2017).
***Idem.***

3) Rebecca Hodge and I (Turowski and Hodge, ESurf 2017) have recently developed a theoretical framework that seems to be ideal to assess the authors' data set. This framework completely separates the issue of cover from sediment transport dynamics. We also defined benchmark cases and describe physical reasons of why a particular cover curve may deviate from these. Your results and the interpretations you propose seem to align with this. And yes, I am obviously biased, but let me know what you think about it.
***We have added references to this publication and touch on the issue of sediment mass vs. sediment load. We also acknowledge the fact that our data should allow determining a probability function as a function of sediment supply and local curvature required for the model you have proposed therein. This would be a very interesting extension to our work and your work. See section 4.4.***

4) There is a great little experimental paper on the interaction of erosion, channel morphology and cover in a meandering channel by Shepherd (Science, 1972; see also the follow up by Shepherd and Schumm, GSAB 1974). This is often overlooked but highly relevant.
***Thanks for pointing this out. We have added references to it in the text.***

Before I make a few more detailed comments specific to the manuscript, I would like to address some common misconceptions about the cover effect that persist in the literature and also shine through here.

1) There is a need to distinguish the relationship between cover and sediment mass on the bed from the relationship of cover with the ratio of sediment supply to transport capacity. Although we claimed in a previous paper (Turowski et al., JGR 2007) that these are equivalent, this is not true. Turowski and Hodge (ESurf 2017) derived a formally correct transformation between these two functions. This point affects the discussion, for example the comparison with the linear cover model (that is typically formulated as a function of the ratio of sediment supply to transport capacity).

*Agreed. We have touched upon this issue on the discussion. See section 4.4.*

2) The exponential cover relation arises when assuming that sediment is randomly distributed on the bed, i.e., each part of the bed has the same probability of deposition. This is true for the relationship between cover and sediment mass, NOT for the relationship of cover with the ratio of sediment supply to transport capacity. The exponential function arises whether one considers the static or the dynamic cover effect, or both combined. There is no relation between the exponential cover relation and the assumption of a dynamic cover effect.

*Please let us know if any part of the revised manuscript still suggests this misconception. We have added a description of the work in Turowski et al. (2007) at the beginning of the discussion and hope is not misleading.  Section 4.1.*

3) The exponential function was derived under the assumptions that flow, sediment supply, bed topography (slope, roughness…) can be considered constant within the area of interest. It is clear that spatial variability should change the functional relationship.

*Yes. We mention this at the beginning of the revised discussion. Section 4.1.*

4) We put forward the dynamic cover effect using the common framework of a subcapacity flow (detachment-limited assumption). In this case, capacity is larger than supply. The river has spare energy for transport and uses this to entrain any stationary sediment. This is the case when assuming that in the control area that is considered, all relevant parameters (hydraulics, supply…) can be considered to be constant (see point 3). In the strict use of these assumptions, a static cover is not possible. This can be resolved by working, for example, in an entrainment-deposition framework (see Turowski, WRR 2009; Shobe et al., GMD 2017; Turowski and Hodge, ESurf 2017).

*Idem.*

Specific comments (page.line)

2.21 This is a misunderstanding. The exponential cover term arises under the assumption that deposition is equally likely on each part of the bed. This is independent of whether there is static or dynamic cover or both. *We believe we have clarified this. See 2.21.*

2.22 Dieter Rickenmann and I presented some field evidence for the dynamic cover effect in a paper in 2009 (Turowski & Rickenmann, ESPL, 2009). *We have added this clarification. See 2.24.*

2.22 The model of Lague is equivalent to the linear / exponential area-based models when working with the mean sediment thickness. *We have added this clarification. See 2.26*

2.25 This paragraph reads like a random selection of statements. The relevance and the main flow of argument are unclear. *We have made small edits to this paragraph. We believe that the information found therein is relevant to provide the reader an up-to-date, quick review of the most recent advances/findings in terms of bedrock-alluvial rivers and the cover effect.*

2.30 …presented… *We have made this change. See 2.32.*

3.9 Shepherd, Science 1972, did some great experiments looking at cover and erosion in a meandering channel. Sadly, his work is often overlooked. There is follow-up paper by Shepherd and Schumm (GSAB 1974). *We have added these references. See 3.10*

3.12 The issue of scale is ok and I agree that this has been insufficiently addressed. Nevertheless, Hodge and Hoey (JGR 2012) and I (Turowski, WRR 2009; Turowski and Bloem, Geodin. Acta 2016) used models to look at cover dynamics from a grain perspective. The assumption in this work is that the appropriate area or length scale to average cover over is much larger than the grain size. It would be easy to assess how much larger it needs with the models developed in the mentioned papers. *We do not make any assumptions regarding the size of the window. We actually conclude suggesting that the maximum window should be as small as possible to capture the fluctuations of alluvial cover. If freely-migrating bedforms are present then the scale is different than if transport is occurring on a particle by particle basis. We think there is still work needed to assess the size of this window. Throughput bedload might need a very different window size than a stream with freely migrating bars or one with forced bars with particles migrating on the edges of them.*

3.14 It is my impression that here two separate issues are confounded. The term 'transient' is ambiguous, because it could mean that a particular patch of cover is created and destroyed over time, or that the sediment in the patch is continuously replaced by new sediment. The relevant point here is not that cover is transient – most model deploy a temporal average or a stochastic notion of a steady state – but that cover patches in a bend may persist in location and thus time-averaged incision varies along and across the channel. This means that, over the long term, the bed is not lowering at the same rate everywhere. *There is no perfect way of referring to alluvial cover. For instance, depending on context, static and permanent cover could be used interchangeably. Transient deposits are an issue of time scales (See section 4.1). The transient cover implied in Sklar and Dietrich (2004) refers to a*

*long term average, but one of the main arguments of the paper is that transient averages lack exactly what you point out, fluctuations of alluvial cover (Fig. 17 in the paper).*

3.16 I do not have the impression that this is not acknowledged by the modelers. **We are not implying that either. We are actually using the references as examples of the issue.**

3.18 This is a good question, and again, there is some work setting up the basics for this. The above-mentioned stochastic models of resolve these aspects explicitly and in a previous paper (Turowski, WRR 2009), I derived probability distributions of cover and erosion rates. Temporal fluctuations in cover are also modelled by Lague (JGR 2010).

The model of Turowski and Hodge (ESurf, 2017) also predicts the temporal evolution of cover. In the latter paper, a mass balance framework is also presented that can easily be extended to look at cover dynamics along and across a channel.

**We have added the references. These papers have a probabilistic framework and we hope that our experiments will contribute to the development and validation of physically based approaches.**

4.3 This notion (transport can happen only on covered portions of the bed) is only true if a dynamic cover effect is assumed. Even then, I find the formulation highly misleading. Clearly, sediment can transported over a clean bedrock bed (think about a single particle driven by water in an otherwise empty flume!), and the notion would imply that there always needs to be a long-stream connected cover. There are many observations suggesting that this is not the case. **Agreed. We have reformulated the text to indicate this. It is however, the assumption that underlies the formulation. It indicates that no matter what the transport capacity is, sediment can only be mobilized from those parts of the bed with sediment. This formulation is not appropriate for the case you mention, which is an end member but similar to the case of throughput bedload. See 4.13.**

4.21 I suggest a more careful formulation here. Yes, in channel bends, there can be stationary deposits. For a given geometry, the location and size of these is dependent on discharge (or the ratio of discharge or capacity to supply). If the term 'mixed bedrock-alluvial channels' indicates a bedrock channel with a partially covered bed, we would need to integrate over the variable discharge to calculate long-term incision rates. During the highest floods, the entire channel may still be stripped of cover. **Agreed.**

5.10 Information on methods and set up is incomplete. What was the size of the sediment (median, distribution)? Was the foam erodible? Why foam? Which type of foam?

How were the foam sections linked (the description suggests that there where sharp steps every 0.5m)? What was the initial pattern of gravel in the flume (spatial distribution, thickness)? Why was this pattern chosen (does the initial distribution affect the results)? Did the concrete completely cover the gravel or did bits stick out, contributing to roughness? Can you give some

more information on the concrete that was used (e.g., the sand fraction may affect the bed roughness)? *Figure 3 has all the grain size distributions. We also added a new supplement to better describe how the bed was built. We have revised the writing to convey the setup in a better way. See Section 2.2 and new supplement S1.*

5.12 How were bedforms averaged out? How were bedforms defined? *We have rewritten this. We extracted transverse slopes from the original bathymetry and fit a relation to the data to represent transverse slope as a function of streamwise location. See supplemental material S1.*

5.16 What is the 'maximum of the foam'? *See supplement S1.*

5.23 Please give information on the model and a reference. *See section 2.3; See 6.2*

5.23 Can you be precise in giving the precision? *See section 2.3; See 6.2*

5.23 Why and how were these locations chosen? Are these line (cross-sections?) or areal scans? *See section 2.3; See 6.3*

5.24 I do not understand the exact protocol here. Why a 4th order polynomial? *We have revised the text and modified it to make it more clear. We also added information in this regard in supplement S1. See section 2.3*

5.25 Why do the authors think that this particular protocol gives and adequate measure of the macro-roughness? *Our approach follows the method proposed by Zhang et al., 2015.*

6.18 If you know the weight, why don't you tell the reader? There is no need to be secretive. *Sorry about this. We missed this detail under the assumption that the information was in Figure 5a. We have added the mass to the legend of figure 5. See 7.1 also. Specific details are included in supplement S1.*

6.18 I have trouble understanding the rationale of this experiment. Please explain more clearly what you intended to do. From the writing, it seems that you wanted to measure the minimum mass of sediment necessary to fully cover the bed, correct? I guess I was confused by the term 'sediment supply ratio'; my first impression was that you wanted to measure sediment supply divided by transport capacity (which would be the typical definition of the term 'sediment supply ratio'). It be would be good to explain the intention of the experiment clearly, including the physical or theoretical reasoning behind it (see Turowski and Hodge, ESurf 2017, for a discussion of why the minimum mass necessary to cover the bed is a useful normalization factor). *We have modified the text to better describe the reasoning behind the experiment. We also added specific details in supplement S1. New text describing this experiment is also found in section 2.5.*

6.22 How was macro-roughness measured? *This is explained in Section 2.3 and supplement S1*

6.25 Why did you think that a single repetition was sufficient to mitigate for human errors? *Because the equipment itself is driven by programmed stepper motors (accurate to*

***1/400th of an inch) and the laser has sub-millimeter precision (250 micro meters). Only source of potential difference between runs one and two was the process of adding the sediment and spreading it over the slab. The results from the two iterations were very similar. See slide 4 in supplement 1.***

8.10 and following. I suggest to use 'persistent' instead of 'permanent' throughout the paper. The cover was not permanent, because it was established throughout the run and removed after it. The dictionary gives a definition of permanent as "lasting or intended to last or remain unchanged indefinitely". ***We have edited the text to use persistent instead of permanent. Thanks for this suggestion.***

8.18 Are these percentages applied to spatial or temporal variance? ***These refers to the temporal variance averaged over the 60 minutes at every pixel in the image. It relates to the maps in Figure 8.***

8.18 This information should go into the method section. ***We prefer to indicate the approach used to differentiate between persistent cover or exposure, and transient alluvial deposits in the specific section of the paper where it is used. It is just a sentence.***

8.22 The legend of the figure should go into the caption. Here, better describe the main features that you want to highlight for the reader. ***We believe this is an issue of preference. We prefer to leave this section as it was written originally. This section of the paper has also been revised anyways to improve readability.***

9.12 Section 3.5 does not give results. The way of calculating erosion potential should be moved to the method section. ***We moved this to the methods section as suggested. See section 2.6.***

9.13 Again, the legend of the figure should be placed in the caption. In the main text, describe the results that you want the reader to see on the figure (e.g., 'A correlates
with B', or 'A shows a maximum at a value of B = XX'). ***This section of the paper has been revised to improve readability. We still include the description of the figure and consider it a matter of preference.***

9.24 dot missing after 'al' ***Dot has been added. P9L18. Thanks.***

10.6 The logistic curve is defined over the space from minus to plus infinity, while the cover is defined between 0 and infinity. Further, we know that (in Fig. 5b), the curve should go through the point 0,0. In a recent paper, Rebecca Hodge and I developed a mathematical framework to deal with such cases (see Turowski and Hodge, ESurf, 2017). ***Yes, mathematically the function can take any value between $\pm\infty$ but alluvial thickness cannot be < 0 (negative). Therefore, physically, the function for alluvial cover is only valid between 0 and infinity. We believe there are no issues here. Might be important in terms of properly bounding a numerical model but otherwise it is common sense.***

11.24 and following. Yes, and this is not really surprising. What are the implications, for example for meander development and channel width? *This belongs to the original discussion, which we modified, based on your comments and those of Christian Braudrick. This section is now in the results section and we have added a comment in this regard. See 11.28.*

Responses to both reviewers submitted at the end of the open discussion period are also included below…

Answers to Jens Turowski

Thanks for your comment to our paper. Both you and Christian have raised the issue of structure and incomplete methods which we will address in a revised version. We need to revisit the literature you have included in your comment before we can address specific comments/issues you have brought up. Be assured we will address all your comments once we have revisited it.

In our response to Christian, we have included some answers to issues you also raised (for example bed construction). We agree that the use of the terms permanent/transient are an open issue. At some point we were referring to the permanent cover as static but decided to avoid the use of 'static' because sediment grains were being transported over these patches as well. Persistent is perhaps a better term. Based on your feedback, and discussions we had internally in terms of the different terms, we will include definitions for each term to clarify how we use it in the paper.

Answers to Christian Braudrick (uploaded at the end of the open discussion period)

Thanks for your comment to our paper. Answers/comments to the issues you have brought up are included below:

    I.    Paper structure: We will work on the paper structure to improve its readability. Both your comments and those of Jens Turowski point in the same direction on this regard.

    II.    Transient cover zone: We discuss this in some sections (e.g. 4.3, 4.4 and 4.5) but perhaps the current structure of the paper is not the best way of conveying the message. We will make sure to highlight the issue better in the revised manuscript.

    III.    Slab experiment: We will make sure to mention this second experiment earlier and highlight its importance. Initial thoughts are that we might add a figure to the main document and extend the supplemental material to thoroughly describe the approach and show images of it.

Regarding your specific questions, here are our thoughts:

1. What is the settling velocity of the particles (calculated or measured). How does that influence the observations?

    Using $D = 1.5mm$, $R = 0.35$, and $v = 1.02e{-}6 m/s^2$, $C1 = 18$, $C2 = 1$
    vs ~ 8.9cm/s (Dietrich) and ~6.9 cm/s (Ferguson and Church). Both values are roughly 2.5 times smaller than those obtained with $R = 1.65$ (22.6cm/s and 16.5cm/s respectively)

    I'm not sure what the effect of heavier sediment would be. My first thought is that it would be harder to push up against the point bar, thus making the permanent cover deposits wider. What do you think?

2. The Shields stress is relatively high (~0.18) does that have an impact on the hop length of the particles, and the locations where impacts are expected? Does the density of the particles affect the hop length?

    Shear stress and particle density definitely affect hop lengths. Similar sand grain would hop less... But we are not focusing on this scale. We cannot see this scale. We can only see sediment patches with our technique. I can't say anything specific about the impacts but our conclusions would still hold with other shear stresses and sediment. The regions of transient alluvial cover will continue to be at the edge of permanent alluvial deposits.

3. The width depth ratio is relatively narrow (~5.5), yet migrating bars are described in the paper. IS there any data on the width depth ratio of bedrock meanders? Are the bars forced bars or free bars (or some other bedform)?, and would the results in a wider channel differ?

    I am not aware of a set of W/H values specific to bedrock meanders but Wohl and David (2008) and Yanites (2010) show W/H values for bedrock rivers. Both include many W/H < 10. Values reported by Wohl and David (2008) have a mean W/H value of 7.7 and a standard deviation of 6.4.
    The experiment shows forced bars (point bars) and free bars migrating over them. I guess it depends on how wide you make the channel. At the apices I think, sediment will continue to travel along the toe of the point bar. At the crossings, however, a wider channel might show other kinds of bars depending on sediment supply.

4.  I would be interested to hear more description of the channel morphology including how deep were the pools and how much did the lateral slope of the bar vary. To some degree, experiments are their own self-contained system, but I couldn't help asking myself how the results would differ if sand or gravel were used rather than walnut shells. Presumably the area of transient cover might be closer to the inner bank with heavier sediment?

    We measured bathymetry for three cover conditions but haven't processed the data to quantify the issues you are interested in. In terms of heavier grains, I think we might see more persistent cover for the same shear conditions. At the apices, the secondary flow will be less successful at pushing grains up against the point bar. Would this actually lead to wider/shallower point bars?

Questions about experimental procedure:
1.  Approximately how thick was the pea gravel?

    It varied from one cross section to another. The mean gravel elevation, measured from the bottom of the channel, was 0.10m. Left and right bank elevations varied following the bathymetry measured in experiments conducted in the same flume using crushed walnut shells in a purely alluvial configuration (Czapiga 2013). The transverse slopes of the bed at streamwise locations were extracted from those experiments and those were used to cut the foam. See Fig. 1 (of this comment). We built a foam 'skeleton' and then filled the channel with pea gravel following the profile defined by the foam cross sections. We then covered everything with a thin layer of concrete.

2.  Was the bed cleaned out between runs? Is the adjustment between runs reflected in any of the results (i.e., how long did it take to adjust the channel morphology)?

    Bed was not cleaned. We started with high volume of sediment (pc79) and removed sediment to conduct runs with lower pc values. Values obtained were not really planned. We just removed enough sediment to achieve a different condition. This lead to pc72 first and pc54 afterwards. After this runs we removed it completely and measured the bare bedrock case (pc00). Following this case we started adding sediment and did the other four conditions (pc19, pc27, pc38 and pc46). We allowed the bed to adjust for at least 8 hours between runs. The 60 minutes we report in the manuscript is after the bed had adapted the new condition. We computed the alluvial cover statistics throughout the transition from one state to another and once it had reached equilibrium we continued measuring. We report only the values once the system had reached equilibrium for each condition.

3.  My memory of walnut shells is that they are pretty angular. Does that affect the experiments at all?

    Yes they are angular and maybe the slab experiments would lead to slightly different results if rounder grains were used (maybe more sediment will be needed to fill the voids in the bedrock bed). Otherwise, our measurements in the Kinoshita flume do not allow focusing on this smaller scale. Figure 3b in the manuscript shows a close up view of the grains for reference.

Other specific comments
Page 5. Lines 10-21. I found the intermingling of "bed material", "alluvium", "bedrock", and "bedrock basement" confusing. Bed material could either refer to the bedrock or

alluvium. I think sticking to bedrock or alluvium would be helpful. I also got confused by the way the description of the artificial bedrock was built starting with the bottom. It might be helpful to start by describing what each component is used for (walnut shells to give the basic channel shape (pools and bars), pea gravel to provide roughness, and concrete to provide strength), then describe how it was built. How thick was the pea gravel?

See Figure 1 (in this comment) for a better description of the bed construction. We will improve the description in the manuscript and incorporate more details to supplement material to thoroughly describe it.

Page 5. Line 19. Is there anything special about the cement mix? What was the ratio of water to sediment (could be useful to future experiments).

No it's just a pre mixed bag (Quickcrete in the US). We followed the bag's instructions for water content. See link: https://www.quikrete.com/pdfs/data_sheet-concrete%20mix%201101.pdf

Section 2.3. What was the scanning interval of the bed, and was the scanner measuring a grid or cross sections (from later in the paper I gathered it was cross sections but I am not sure) ?

We did scan cross sections. The laser was attached to a stepper motor that moved in 0.42 mm displacements along the cross section.

The slope was measured with point gages from 9 m to 21 m. These spots are just before the apex of one bend and after the apex of another bend. Often the water surface elevation can vary based on position in the bend as water backs up behind the bend apex. Did this occur in these experiments? If so, are these slope measurements representative of the overall slope.

Figure 4 (in the manuscript) shows how different the overall slope (measured between upstream and downstream tanks) was with respect to the slope measured in the middle portion of the flume. We decided to measure close to our region of interest but do not know if the measurements were affected by backwater effects as you suggest.

Page 10, lines 5-15. Does the thickness of the alluvial cover depend on the shape of the particles?

Perhaps rounder particles would build slightly shallower deposits but the areal alluvial cover trends we focus in would tend to be similar.

Figure 10. Please define area ratio in the figure legend.

Ok. We will do this.

Figure 11. Can we quantify the variability of cross sectional alluvial cover relative to reach average cover using standard deviation, or just a range?

Figure 13 shows box plots of local pc (normalized with reach-averaged values). This shows the ranges observed locally.

Figure 13. I found myself curious how symmetrical the results were and wondered why the cover was higher upstream than downstream. How did cover vary as a function of

the absolute value of curvature? Presumably the curvature upstream and downstream matters as well as local curvature.

To see how the trend in Figure 13 looks with respect to absolute curvature use both the true and negative curvature signals. Follow the true signal from CS10 to CS 12.8. Then 'hop on' the negative signal and follow it until CS 17.8. At this point, 'jump back' onto the true curvature signal. We decided to include true and negative curvature signals because the sharp changes in the absolute curvature signal at the crossings did not look great. We believe that using both signals seems to better show the trend.

As for more cover upstream than downstream it might suggest that the curvature sign also has an influence, and as you suggest, the curvature leading to a given point. One way to know would be to run the experiments and focus on more than one bend and see if the trends repeat themselves one bend after another. This of course can't be done now but is an interesting thing to try to answer with numerical modelling

[revised manuscript text omitted]

---

## Referee Report (RR1)

The paper by Fernandez et al reports on a very interesting set of experiments exploring bedrock erosion in a sinuous channel. Their novel approach and clever measurements make this paper a crucial addition to the bedrock erosion literature. The videos that will be included in the SI are fascinating. Several notable findings, including that the areas of transient and permanent cover were of similar magnitude will be useful to develop a bedrock erosion theory for sinuous channels. The authors responded to the previous comments including fixing the structure of the paper, beefing up some of their citations as pointed out by Jens, and addressed minor comments throughout the document. The comments below are very minor, and I do not think another review would be necessary once these comments are addressed.

The details about the transition between experiments were include in the response to comments, but not included in the revised paper or SI that I could see. To avoid any confusion, I think those details should be included in the methods section.

A sentence or two about the resolution of the camera in the SI or paper would be helpful.

Page 6. The discussion of the  polynomial in the SI was clarifying and the SI should be referenced in the text where the polynomial was discussed.

Page 6, Line 20. A sentence about the pixel size of the camera would be helpful to interpret the results. If the pixel size is larger than the particle than the bed  Presumably particles are mostly grouped, was the threshold to detect the presence of sediment set to pick up if a single particle fell into the pixel or is it likely that more than a single particle was necessary for Matlab to record the point as containing sediment? This may not matter since it sounds like the particles were moving in bedforms, but it might be worth addressing, and least in the SI.

Page 7, Line 7 and Figure 5a,b. Shouldn't figures 5a and 5b have two sets of data on them since the experiments were repeated? I am very surprised that the experiments were so repeatable. I think the authors misunderstood Jens' comments about human error. The most likely sources of human error are evenly distributing the particles.

Page 8 Line 6, Figure 2 shows that the bedload trap is downstream not upstream.

Page 9. The organizational issues have mostly been dealt with, with the exception of the description of figure 6 in Lines 11-14, which comes between the description of Figure 5 and the discussion of Figure 5.

Page 10. Equations 9 and 10. These equations feel like the should come before the previous paragraph to be immediately below the paragraph that describes them.

Page 11 and Figure 10. I spent a lot of time thinking about this figure and the discussion in the text. I wondered whether the small increases and decreases in persistent and transient cover were repeatable or if they were just scatter in the data. I think its just as likely that its scatter as it is due to a trend in the data. Another explanation is that equilibrium had yet to be reached when the data recording began. Figure 7 shows that the cover varied temporally in Pc=0.78, 0.72, 0.46,

0.38, and 0.27 (where there was a increasing cover trend). It is really interesting that the persistently covered area and transiently covered area are similar. What a cool finding.

Page 11, Line 20. Why would the depth be constant? If the slope and roughness are changing you would think that the depth and velocity would adjust. I don't recall water surface (or velocity) measurements. If the slope is changing isn't the depth changing too?

Page 15, bottom. Did the authors observe particles striking the bank, or is that inferred?

Page 16, Lines 21-26. Presumably the distribution of the grain size and roughness matter too, which would likely be larger in the field? Also, there is a lot in "hydraulic conditions".

Table 2. Inferring the middle reach slopes for pc54-79 to two decimal points seems like too much. It is hard to justify 2 decimal points, and looking at the table, it seems like there would be a lot of scatter in the relationship between overall slope and middle bend slope. The more I think about this the more I think they shouldn't be included at all. The slope measurement with two local data points is probably not very robust anyway.

---

## Author Response (AR2)

01 August 2019

Dr. Eric Lajeunesse
Associate Editor
Earth Surface Dynamics

Dear Eric:

We have made minor edits to the manuscript following the recommendations that Jens, Christian and yourself have made. Thanks for your patience while handling the manuscript and for your feedback. The following pages contain all feedback provided and our reply (***bold italics font***) to each particular point brought up. The 'Track Changes' version of the manuscript is attached at the end of this document. References to page, line in our replies correspond with the 'Track Changes' version.

We look forward to the publication of this manuscript and to future submissions to ESurf.

Sincerely,

Roberto Fernández
On behalf of all authors

**ESurf – Round of Reviews No. 2 – Minor Revisions**

Associate Editor Decision: Publish subject to minor revisions (review by editor) (20 Jul 2019) by Eric Lajeunesse

Comments to the Author:

Dear Roberto Fernandez,

Thank you for submitting a revised version of your manuscript entitled "Experiments on patterns of alluvial cover and bedrock erosion in a meandering channel" to ESurf. I have received the reports of the two referees. Both of them feel that the manuscript is much improved. Yet they report a detailed list of small issues which need to be resolved before the paper can be published in ESurf. In particular, I regret that several methodological points answered in the rebuttal are not included in the new version of the manuscript.

***Thanks Eric. We have added a few more details and relevant references to the supplemental material throughout the text to make sure all information regarding methodological issues is covered. See Sections 2.2, 2.3, 2.4, 2.6.***

Reading the new version of the manuscript, I have noted that the logarithmic term log( qs/qbt ) in equation (8) diverges when the sediment supply ratio tends towards 0. This prediction, likely inconsistent with Physics, suggests that equation (8) does not describe the small sediment flux limit. If so, this fact should be clarified and discussed carefully. I also recommend you to use a logarithmic scale for the horizontal axis of Figure 5c, as this is the only way to evidence unambiguously a logarithmic dependency of the areal fraction of alluvial cover pc with the sediment supply ratio qs/qbt.

***Thanks for pointing this out. We modified figure 5c to include a logarithmic horizontal axis and have included a statement saying that Eq. 8 does not describe the small sediment flux. P10.L1-2. Figure 5.***

I encourage you to submit a suitably revised version of your manuscript taking into account the remaining issues raised by the reviewers and myself. Upon submission, we will need to receive a response to reviewer file that lists each of the comments and describes how the manuscript has been modified (or not) in response to those comments.

I look forward to receiving your revised manuscript.

Sincerely yours,

Eric Lajeunesse

Reviewer 1 – Jens Turowski

Dear authors:

Thanks for the revised manuscript. I read it with interest. The paper is somewhat improved and the methods are largely clear now. Nevertheless, I still find inaccuracies especially in the introduction, and the research gap and aims are not cleanly set up. The material presented in the introduction needs to be focused and stream-lined.

I appreciate that you picked up some of my previous suggestions in the discussion. Yet, I am a little surprised that some of the most obvious similarities are glanced over. As an example, in the Turowski and Hodge paper, we derive a cover equation with a logarithmic shape somewhat similar to what you fit to the experimental data (see eq. 27 and 31 in that paper). I wonder why this is not discussed.

There is also a mixture of discussion material in the results. Please look out for this and move / rewrite accordingly. I might have overlooked something, but it seems to me that some of the methodological points were answered in the rebuttal, but didn't make their way into the manuscript (e.g., properties of the cement, asked both by me and Christian Braudrick). Please review and amend this.

Finally, the 6 points in the conclusion do not directly pertain to the four points that are said to be addressed by the study at the end of the introduction. It would be good to rewrite this to have a direct relationship.

***Thanks Jens. We have made edits to the manuscript following your recommendations. Specific aspects are mentioned below. To answer your general concerns:***

- ***We are happy with the introduction we have written.***
- ***We have added an explicit reference to Eqs. 27 and 31 in Turowski and Hodge(2017) where we introduce Eq. 8 in our manuscript.***
- ***We have moved the 'discussion material' to the discussion and changed the order of the three last figures (and references to them in the text).***
- ***Our 6 conclusions pertain to all four points in the introduction. Some conclusions pertain to more than a single aspect in the introduction. It was not our objective to provide a single conclusion per issue raised in the introduction.***

2.8 …Gilbert did not specifically use…

***We removed the year.  P2L8***

2.14 only the 2001 paper includes experimental work

***We removed the 1998 reference. P2L14***

2.15 The description here relates just to the cover effect. The tools effect is parameterized as a linear dependence on sediment supply in the saltation-abrasion model.

***We edited the paragraph to indicate this. P2.15-17***

2.16 The saltation-abrasion model…

***We have made this edit. P2.L17***

2.25 Lague's model is not for bedrock incision, but for bedrock channel morphodynamics

*We have made this clarification. P2.L26.*

2.26 Misleading: his cover model is equivalent as described, the context suggests that you mean the morphodynamics model.

*We have made this clarification. P2.L27*

2.28 and following: the paragraph features a string of loosely connected statement (jumping from a brief description of Zhang's cover model, to Turowski's 1D model of steady state channel morphology to Mishra's experiments) and lacks a clear line of argument. The references are treated at different level of detail; a symmetry is lacking. The point of the introduction should not be to list all previous cover models and their applications, but provide an argument that leads to the research gap and question.

*We provide a more detailed description of recent advances because they are relevant to our work. We then say that even these advances are lacking certain aspects and introduce our research questions.*

3.4-5 I somewhat disagree with this statement of the research gap, because in my mind it confounds two unconnected problems. Let me explain this briefly. When we think about cover, we have two problems to solve. Consider a particular control area on the bed (in a discrete model, this would be a pixel, in a continuous formulation an infinitesimal area element). 1) For a given amount of sediment mass or volume, we need to know how this is distributed to give a particular cover fraction. As a concrete example, if we have, say, 1/100 of a cubic meter of sediment deposited on an area of 1m2, we could have a complete cover of 1cm thickness of the entire area, or we could cover half of the area by a 2cm thick layer. Or there are an infinity of other possibilities to distribute this sediment. Therefore, we need a relationship between cover fraction and sediment mass (or volume). The point of the exponential cover function of Turowski et al. 2007 and Turowski 2009, and of the P function of Turowski and Hodge 2017 is to provide such a relationship. 2) We need to know how much sediment deposits in a given area element (the total mass or volume per area element of the bed). This is done by relating deposition and erosion to hydraulics. In general, this is done either using a framework based on the Exner equation (e.g. Zhang's or Inoue's work) or on the entrainment-deposition framework (e.g., Turowski and Hodge 2017, Shobe et al. 2017).

*We don't understand the difference between 1) and 2). If you know how a given mass of sediment is distributed over a bed area, leading to a cover fraction (point 1) you immediately know how much sediment deposits in a given area element (point 2*). **How are these different/unconnected? Perhaps we can discuss this further when we cross paths.**

3.7 There is also earlier work by Nelson and Seminara (2011 and 2012).

*We have added the references to their work. P2.L4, P2.L9*

3.8 Again, here is a confusion between morpho-dynamic models and models of bedrock incision. As an example, the saltation-abrasion model is a model of bedrock incision; it predicts erosion rate (that is volume removed per time and per bed area) as a function of hydraulics, and sediment properties and transport rate. It has been used in 0D/1D morphodynamics models for example by Sklar and Dietrich 2006, by Zhang et al., or by Turowski 2018. Similarly, the stream power / shear-stress model is a bedrock incision model, which has been used in a number of morphodynamics frameworks (e.g., Wobus et al. 2006, Stark, 2006).

*We have added 'morphodynamics models' to clarify. P3.L9-10*

3.9-11 What about the experiments by Friedl (chapter 7 of https://www.ethz.ch/content/dam/ethz/special-interest/baug/vaw/vaw-dam/documents/das-institut/mitteilungen/2010-2019/245.pdf, mentioned in one of my previous comments)?

*The experiments of Friedl look at the effects of sediment supply over an armoured bed. Although there are connections between the cover patterns seen therein and our work, their focus was not on the cover effect. The citations we included in this paragraph are specific to the cover effect in bedrock rivers.*

3.10 I still think the formulation is misleading. The reply in the rebuttal does not convince me (it is not the assumption of the model that sediment can only be transport on covered parts of the bed). Maybe it would be good if you make the assumptions underlying your interpretation of the model explicit, and more clearly explain your line of thinking.

*Line 4.10.*

*If the transport capacity corresponds to a fully alluviated bed, the term qbs = pc\*qbt implies that sediment transported in the reach is only available in those areas of the bed that are covered with sediment. It doesn't imply that sediment is only transported over covered parts but rather that sediment can only be mobilized from alluviated areas.*

*P4.L10*

3.13 Maybe it would be helpful to define 'throughput load' here.

*Line 4.13.*

*We have added a definition. P.4L13-14.*

3.14 Again, this is inaccurate. Yes, some models, notably the linear cover model, are based on temporally representative values (essentially using long-term 'effective' values for hydraulics and sediment supply). Still, sediment supply and transport capacity could be written as functions of time. In that way, one can model temporally evolving cover. This approach neglects the response time of cover to changing conditions, i.e., cover is always adjusted to the boundary conditions and any lags are neglected. The mass/volume based formulations are more general, one could write mass/volume as a function of time. The crux is to couple these cover models to proper hydraulic and sediment transport models. And even this has been done in a handful of papers (most of which you cite earlier). Finally, in the Turowski and Hodge 2017, we devoted an entire section to response time scales, including analytical solutions and explicit formulations of response time scales, and a discussion of leads and lags, going far beyond the 'loose definition' that you claim to exist to date. Regarding the spatial time scale, there are papers that upscale from the grain scale (which I already pointed out in the last round of reviews). Again, here the problem is related to appropriate length scales of hydraulics, rather than cover.

*We have changed 'only loose' to 'different'. P3. L16*

3.17 I still think you need to supply a clear definition of what you mean by transient cover.

*We decided to leave this open on purpose and then discuss the issue in section 4.1.*

3.20-24 Again, I point to the Turowski and Hodge paper. The model provided therein for sediment transport is physically based and can deal with temporal cover fluctuations (see their figures 10, 11 for examples of temporal evolution of cover in generic situations and 14 for an example calculation

for a real flood event). Another model (that I did not mention in my previous review) is the one of Johnson, JGR 2014. He deals with feedbacks between cover and bed roughness.

*We have edited the text and now it says that 'physical measurements' as opposed to 'physically based approaches'. P3.L24.*

3.25 From an experimental point of view the question is fair. However, I suggest to state that there are models that predict a much wider range of behaviors. See for example the cellular automaton of Hodge and Hoey 2012 or the simulations of Aubert et al. (ESurf 2016). The P-function framework of Turowski and Hodge is sufficiently flexible to be able to describe all of these observations.

*We have edited the text to include this observation. P3. L26, 28, 29-30.*

3.28 It has also been shown to not be an acceptable approximation in previous work (incidently, also in some of the cases described in the cited paper).

*That is why the text said 'different circumstances' not 'all'. We edited to 'certain circumstances' to avoid confusion. P3. L28*

4.7 why 'alternative'? The saltation-abrasion model evaluates to the same equation (2) once all the terms in (1) are evaluated. We do not show how Eq. (1) becomes Eq. (2).

We *have removed 'alternative' and just call it a closely related equation.  P4. L6-7*

4.13 I do not see this readily. If we view p_0 and p_c as a function of time and space, the equation is still valid. (And this is exactly what is done in the following paragraph.) See also my comment to 3.14.

*The argument being made is about the values of pc and po which will always be zero or one in the case of throughput bedload. Even if you average in space and/or time, the bed will never have alluvial cover deposits for as long as the bedload enters and leaves the reach without depositing (i.e. particles continuously saltate, roll or slide).*

4.14 Full stop instead of comma after sense.

*Thanks, this was a typo. We have edited it.  P4.L11.*

4.15 grammar / sentence unclear (P_0 is this but instead that).

*We have edited the text slightly to improve readability. P4.L16, L17.*

14.12 Slightly inaccurate; the assumption is of uniformity within the area of consideration. One could divide the bed in multiple cells where this assumption is (approximately) valid and in this way model cross-stream variations.

Although in my previous papers, I have dealt with a control area with homogenous conditions, it would be straight-forward to expand this to full 2D models, by placing a number of homogenous control areas in the long- and in the cross channel direction (building up a grid that represents the bed). Then, the only thing that needs to be added is the long- and cross-stream sediment mass balance as a function of hydraulics, i.e., we need to specify how much sediment moves from a given cell into the neighboring cells. This is straight-forward (though not necessarily easy) because the tools to do this are available (for example, the Nelson and Seminara papers, or other work that deals with 2D resolved sediment transport in alluvial rivers). The question again becomes of how large/small can the considered cells become. We want to make them as small as possible to properly

resolve the hydraulic variations, but large enough so that numerical computations are feasible, and that we can treat sediment as a continuous mass and do not need to deal with granular effects.

*We have modified 'channel width' to 'considered area of unspecified dimensions'. P15.L3-4*

4.19 It is not an arbitrary function. 'Arbitrary' means that it does not matter which function is chosen. Instead, there are some clear constraints on the function, both from fundamental logic / definitions (i.e., cover cannot be smaller than 0) and from observations.

*We have removed the word 'arbitrary'. P4.L20*

4.20 What are your arguments to judge this as the 'simplest realistic form'? What observations does your realism hang on? Not sure whether I would call a discontinuous function realistic…

*We have edited the text and refer to them as 'commonly used functions'. P4.L21*

4.27 The statement '…and channel geometry would not change over time' does not follow from what has been said before. Clearly, channel morphology changes if, say, only the right side of the channel bed erodes and not the left side. I would even argue the other way round: as long as there is erosion somewhere (partially covered bed) persistent cover necessarily leads to an adjustment of channel geometry!

*If pc = 0 or pc = 1 erosion would always be zero according to the relations presented above. Therefore, channel geometry would indeed not change in time.*

4.28 Again, this is not necessarily true. You may have patches of cover moving through and still adjust the channel (for example if wall erosion rates are much high than bed erosion rates). This is because erosion rate is not only dependent on the cover effect, but also on the tools effect (tools concentration may vary, as may the impact rate for a given concentration, or there may be a lateral sorting of grain sizes).

*We changed 'cross section' to reach and added a reference to Sklar and Dietrich to avoid confusion. P4. L28-30.*

5.1 Better: may form – not all meander bends necessarily have point bars at all times. Also consider temporal variation of floods – we do not really know whether point bars persist throughout the largest floods.

*We have added the word 'may'. Thanks for this observation. P5.L1.*

5.2 …breaks down…

*Refers to assumptions. 'break down' is correct.*

5.18 What point cloud?

*It referred to the bathymetry. We have edited the text to make it clear. P5.L17-18.*

5.31 The term macro-roughness appears here for the first time pertaining to the experiment. What do you mean by it?

*We have added our definition of bedrock macro-roughness when it appeared for the second time… Thanks for pointing this out. We have moved the definition to this paragraph. P.5.L31.*

9.20 Eq. (8) – compare to the results of Turowski and Hodge, e.g., eq. 27 and 31 in their paper. This gives a similar logarithmic function in the cover domain, where the exponential term is approximately zero.

*We have added a reference to this paper and the specific equations. P10.L1-3.*

Aubert et al. (ESurf 2016) also predict a similar function from their models.

*We also added a reference to this paper. P10.L3-4*

9.31 I am still not convinced by the use of the logistic curve. For qs/qt=0, this gives finite cover, which is unphysical. It is not difficult to find an equation that honors that condition and otherwise has similar poperties.

*We have made a clarification on this regard. The model of Zhang et al (2018, 2015) and others based on it (e.g. Inoue and colleagues) use the alluvial cover to macro-roughness ratio between a characteristically low (e.g. 0.05) and a characteristically high (e.g. 0.95) value to avoid singularities or unphysical results. P10.L23-24.*

11.30 this is an interpretation that should be placed into the discussion section

*We have moved it to the discussion. P15.L20-21, 24-26.*

13.12-26 this is discussion material

*Idem. P16.L30-31 and P17.L1-15.*

13.31-14.3 discussion material

*We left this here because it actually serves as the transition point between the results and discussion section.*

14.28 no comma after 'above'

**This was a typo. We have removed the comma. P15.L19.**

16.7 ...and Nelson and Seminara 2011

*Yes! We have added this reference. P17.L23*

17.2 They showed that energy can transferred, not that erosion is possible. Erosion was not investigated in this study.

*Thanks. We have edited the wording and mention energy instead of erosion. P18.L18-19*

17.15 citation incorrect, should be Turowski and Hodge 2017.

*Thanks. We have modified it. P18.L32*

Reviewer 2 – Christian Braudrick

The paper by Fernandez et al reports on a very interesting set of experiments exploring bedrock erosion in a sinuous channel. Their novel approach and clever measurements make this paper a crucial addition to the bedrock erosion literature. The videos that will be included in the SI are fascinating. Several notable findings, including that the areas of transient and permanent cover were of similar magnitude will be useful to develop a bedrock erosion theory for sinuous channels. The authors responded to the previous comments including fixing the structure of the paper, beefing up some of their citations as pointed out by Jens, and addressed minor comments throughout the document. The comments below are very minor, and I do not think another review would be necessary once these comments are addressed.

The details about the transition between experiments were include in the response to comments, but not included in the revised paper or SI that I could see. To avoid any confusion, I think those details should be included in the methods section.

***We have added more details regarding the transition between runs and more references to the supplemental material to make sure that the methods are clear. P7.L26-27, L30-31, P8.L1-2.***

A sentence or two about the resolution of the camera in the SI or paper would be helpful.

***We added this information in section 2.4. P6. L21-24***

Page 6. The discussion of the polynomial in the SI was clarifying and the SI should be referenced in the text where the polynomial was discussed.

***We have added the reference to the supplemental material. P6.L8-9.***

Page 6, Line 20. A sentence about the pixel size of the camera would be helpful to interpret the results. If the pixel size is larger than the particle than the bed Presumably particles are mostly grouped, was the threshold to detect the presence of sediment set to pick up if a single particle fell into the pixel or is it likely that more than a single particle was necessary for Matlab to record the point as containing sediment? This may not matter since it sounds like the particles were moving in bedforms, but it might be worth addressing, and least in the SI.

***We added this information in section 2.4. P6.L21-24.***

Page 7, Line 7 and Figure 5a,b. Shouldn't figures 5a and 5b have two sets of data on them since the experiments were repeated? I am very surprised that the experiments were so repeatable. I think the authors misunderstood Jens' comments about human error. The most likely sources of human error are evenly distributing the particles.

***We have added both sets of data (circles and squares in Figs. 5a, 5b). See figure 5.***

Page 8 Line 6, Figure 2 shows that the bedload trap is downstream not upstream.

***We have clarified the text. P.8.L20-21.***

Page 9. The organizational issues have mostly been dealt with, with the exception of the description of figure 6 in Lines 11-14, which comes between the description of Figure 5 and the discussion of Figure 5.

***We have moved the paragraph following this recommendation. Thanks. P11.L1-4.***

Page 10. Equations 9 and 10. These equations feel like the should come before the previous paragraph to be immediately below the paragraph that describes them.

***We have moved the equations following this recommendation. Thanks. P10.L20-21***

Page 11 and Figure 10. I spent a lot of time thinking about this figure and the discussion in the text. I wondered whether the small increases and decreases in persistent and transient cover were repeatable or if they were just scatter in the data. I think its just as likely that its scatter as it is due to a trend in the data. Another explanation is that equilibrium had yet to be reached when the data recording began. Figure 7 shows that the cover varied temporally in Pc=0.78, 0.72, 0.46, 0.38, and 0.27 (where there was a increasing cover trend). It is really interesting that the persistently covered area and transiently covered area are similar. What a cool finding.

***We definitely need to look at a lot of data not included in this manuscript which relates to the conditions between one state and another. We also have data for hours and hours of the same run which would allow determining if it's scatter or a trend.***

Page 11, Line 20. Why would the depth be constant? If the slope and roughness are changing you would think that the depth and velocity would adjust. I don't recall water surface (or velocity) measurements. If the slope is changing isn't the depth changing too?

***We have adjusted the text to clarify this point. The no-flow water depth was kept constant but it certainly changes based on the amount of sediment in the flume and the percent of cover. The water surface slope changed accordingly. We have a paper in review in JHR looking at the instantaneous water surface elevation changes based on the eTape readings as alluvial cover changes in the flume. P12.L11-13.***

Page 15, bottom. Did the authors observe particles striking the bank, or is that inferred?

***This is inferred. We introduce this set of arguments with 'Our results suggest'. P16.L27.***

Page 16, Lines 21-26. Presumably the distribution of the grain size and roughness matter too, which would likely be larger in the field? Also, there is a lot in "hydraulic conditions".

***We edited the text to include 'grain size'. Agreed. P18.L7.***

Table 2. Inferring the middle reach slopes for pc54-79 to two decimal points seems like too much. It is hard to justify 2 decimal points, and looking at the table, it seems like there would be a lot of scatter in the relationship between overall slope and middle bend slope. The more I think about this the more I think they shouldn't be included at all. The slope measurement with two local data points is probably not very robust anyway.

***We removed a decimal point. Table 2.***

[revised manuscript text omitted]

$$\underline{\qquad\qquad\qquad\qquad\qquad\qquad\qquad\qquad} \tag{9}$$

$$\underline{\qquad\qquad\qquad\qquad\qquad\qquad\qquad\qquad} \tag{10}$$

The function in Eq. 10 is valid between a characteristically low (e.g. 0.05) and a characteristically high (e.g. 0.95) value of      to avoid unrealistic cover values (Zhang et al. 2018). It is likely that the steepness and midpoint value are associated with some measure of the grain size distribution of the alluvium and the macro-roughness height of the bedrock. In the case of the bedrock and alluvium (Fig. 3a) used in this study, the steepness value corresponds to                     and the mid-point value corresponds to                     . This issue merits further investigation so as to define appropriate relations to calculate the steepness and mid-point value of the sigmoid curve for implementation in numerical models. We discuss the issue of alluvial thickness and alluvial cover further in section 4.4.

$$\underline{\qquad\qquad\qquad\qquad\qquad\qquad\qquad\qquad} \tag{9}$$

$$\underline{\qquad\qquad\qquad\qquad\qquad\qquad\qquad\qquad} \tag{10}$$

[revised manuscript text omitted]